# Online Prediction with Selfish Experts

**Tim Roughgarden**
Department of Computer Science
Stanford University
Stanford, CA 94305
tim@cs.stanford.edu

**Okke Schrijvers**
Department of Computer Science
Stanford University
Stanford, CA 94305
okkes@cs.stanford.edu

## Abstract

We consider the problem of binary prediction with expert advice in settings where experts have agency and seek to maximize their credibility. This paper makes three main contributions. First, it defines a model to reason formally about settings with selfish experts, and demonstrates that "incentive compatible" (IC) algorithms are closely related to the design of proper scoring rules. Second, we design IC algorithms with good performance guarantees for the absolute loss function. Third, we give a formal separation between the power of online prediction with selfish versus honest experts by proving lower bounds for both IC and non-IC algorithms. In particular, with selfish experts and the absolute loss function, there is no (randomized) algorithm for online prediction—IC or otherwise—with asymptotically vanishing regret.

## 1 Introduction

In the months leading up to elections and referendums, a plethora of pollsters try to figure out how the electorate is going to vote. Different pollsters use different methodologies, reach different people, and may have sources of random errors, so generally the polls don't fully agree with each other. Aggregators such as Nate Silver's FiveThirtyEight, and The Upshot by the New York Times consolidate these different reports into a single prediction, and hopefully reduce random errors.[1] FiveThirtyEight in particular has a solid track record for their predictions, and as they are transparent about their methodology we use them as a motivating example. To a first-order approximation, they operate as follows: first they take the predictions of all the different pollsters, then they assign a weight to each of the pollsters based on past performance (and other factors), and finally they use the weighted average of the pollsters to run simulations and make their own prediction.[2]

But could the presence of an institution that rates pollsters inadvertently create perverse incentives for pollsters? The FiveThirtyEight pollster ratings are publicly available.[3] They can be interpreted as a reputation, and a low rating can negatively impact future revenue opportunities for a pollster. Moreover, it has been demonstrated in practice that experts do not always report their true beliefs about future events. For example, in weather forecasting there is a known "wet bias," where consumer-facing weather forecasters deliberately *over*estimate low chances of rain (e.g. a $5\%$ chance of rain is reported as a $25\%$ chance of rain) because people don't like to be surprised by rain [Bickel and Kim, 2008].

These examples motivate the development of models of aggregating predictions that endow agency to the data sources.[4] While there are multiple models in which we can investigate this issue, a natural candidate is the problem of prediction with expert advice. By focusing on a standard model, we abstract away from the fine details of FiveThirtyEight (which are anyways changing all the time), which allows us to formulate a general model of prediction with experts that clearly illustrates why incentives matter. In the classical model [Littlestone and Warmuth, 1994, Freund and Schapire, 1997], at each time step, several experts make predictions about an unknown event. An online prediction algorithm aggregates experts' opinions and makes its own prediction at each time step. After this prediction, the event at this time step is realized and the algorithm incurs a loss as a function of its prediction and the realization. To compare its performance against individual experts, for each expert the algorithm calculates what its loss would have been had it always followed the expert's prediction. While the problems introduced in this paper are relevant for general online prediction, to focus on the most interesting issues we concentrate on the case of binary events, and real-valued predictions in $[0, 1]$. For different applications, different notions of loss are appropriate, so we parameterize the model by a loss function $\ell$. Thus our formal model is: at each time step $t = 1, 2, \ldots, T$:

1. Each expert $i$ makes a prediction $p_i^{(t)} \in [0, 1]$, representing advocacy for event "1."
2. The online algorithm commits to a probability $q^{(t)} \in [0, 1]$ as a prediction for event "1."
3. The outcome $r^{(t)} \in \{0, 1\}$ is realized.
4. The algorithm incurs expected loss $\ell(q^{(t)}, r^{(t)})$, each expert $i$ is assigned loss $\ell(p_i^{(t)}, r^{(t)})$.

The standard goal in this problem is to design an online prediction algorithm that is guaranteed to have expected loss not much larger than that incurred by the best expert in hindsight. The classical solutions maintain a weight for each expert and make a prediction according to which outcome has more expert weight behind it. An expert's weight can be interpreted as a measure of its credibility in light of its past performance. The (deterministic) Weighted Majority (WM) algorithm always chooses the outcome with more expert weight. The Randomized Weighted Majority (RWM) algorithm randomizes between the two outcomes with probability proportional to their total expert weights.

The most common method of updating experts' weights is via multiplication by $1 - \eta\ell(p_i^{(t)}, r^{(t)})$ after each time step $t$, where $\eta$ is the learning rate. We call this the "standard" or "classical" version of the WM and RWM algorithms.

The classical model instills no agency in the experts. To account for this, in this paper we replace Step 1 of the classical model by:

1a. Each expert $i$ formulates a belief $b_i^{(t)} \in [0, 1]$.
1b. Each expert $i$ reports a prediction $p_i^{(t)} \in [0, 1]$ to the algorithm.

Each expert now has two types of loss at each time step — the *reported loss* $\ell(p_i^{(t)}, r^{(t)})$ with respect to the reported prediction and the *true loss* $\ell(b_i^{(t)}, r^{(t)})$ with respect to her true beliefs.[5]

When experts care about the weight that they are assigned, and with it their reputation and influence in the algorithm, different loss functions can lead to different expert behaviors. For example, for the quadratic loss function, in the standard WM and RWM algorithms, experts have no reason to misreport their beliefs (see Proposition 8). This is not the case for other loss functions, such as the absolute loss function.[6] The standard algorithm with the absolute loss function incentivizes extremal reporting, i.e. an expert reports 1 whenever $b_i^{(t)} \geq \frac{1}{2}$ and 0 otherwise. This follows from a simple

derivation or alternatively from results in the property elicitation literature.[7] This shows that for the absolute loss function the standard WM algorithm is not "incentive-compatible" in a sense that we formalize in Section 2. There are similar examples for the other commonly studied weight update rules and for the RWM algorithm. We might care about truthful reporting for its own sake, but additionally the worry is that non-truthful reports will impede our ability to get good regret guarantees (with respect to experts' true losses).

We study several fundamental questions about online prediction with selfish experts:

1. What is the design space of "incentive-compatible" online prediction algorithms, where every expert is incentivized to report her true beliefs?

2. Given a loss function like absolute loss, are there incentive-compatible algorithms with good regret guarantees?

3. Is online prediction with selfish experts strictly harder than in the classical model with honest experts?

**Our Results.** The first contribution of this paper is the development of a model for reasoning formally about the design and analysis of weight-based online prediction algorithms when experts are selfish (Section 2), and the definition of an "incentive-compatible" (IC) such algorithm. Intuitively, an IC algorithm is such that each expert wants to report its true belief at each time step. We demonstrate that the design of IC online prediction algorithms is closely related to the design of strictly proper scoring rules. Using this, we show that for the quadratic loss function, the standard WM and RWM algorithms are IC, whereas these algorithms are not generally IC for other loss functions.

Our second contribution is the design of IC prediction algorithms for the absolute loss function with non-trivial performance guarantees. For example, our best result for deterministic algorithms is: the WM algorithm, with experts' weights evolving according to the spherical proper scoring rule (see Section 3), is IC and has loss at most $2 + \sqrt{2}$ times the loss of best expert in hindsight (in the limit as $T \to \infty$). A variant of the RWM algorithm with the Brier scoring rule is IC and has expected loss at most 2.62 times that of the best expert in hindsight (also in the limit, see Section 5).

Our third and most technical contribution is a formal separation between online prediction with selfish experts and the traditional setting with honest experts. Recall that with honest experts, the classical (deterministic) WM algorithm has loss at most twice that of the best expert in hindsight (as $T \to \infty$) [Littlestone and Warmuth, 1994]. We prove in Section 4 that the worst-case loss of every (deterministic) IC algorithm, and every (deterministic) non-IC algorithm satisfying mild technical conditions, is bounded away from twice that of the best expert in hindsight (even as $T \to \infty$). A consequence of our lower bound is that, with selfish experts, there is no natural (randomized) algorithm for online prediction—IC or otherwise—with asymptotically vanishing regret.

Finally, in Section 6 we show simulations that indicate that different IC methods show similar regret behavior, and that their regret is substantially better than that of the non-IC standard algorithms, suggesting that the worst-case characterization we prove holds more generally.

**Related Work.** We believe that our model of online prediction over time with selfish experts is novel. We next survey the multiple other ways in which online learning and incentive issues have been blended, and the other efforts to model incentive issues in machine learning.

There is a large literature on prediction and decision markets (e.g. Chen and Pennock [2010], Horn et al. [2014]), which also aim to aggregate information over time from multiple parties and make use of proper scoring rules to do it. However, prediction markets provide incentives through payments, rather than influence, and lack the feedback mechanism to select among experts. While there are strong mathematical connections between cost function-based prediction markets and regularization-based online learning algorithms in the standard (non-IC) model [Abernethy et al., 2013], there does not appear to be any interesting implications for online prediction with selfish experts.

There is also an emerging literature on "incentivizing exploration" in partial feedback models such as the bandit model (e.g. Frazier et al. [2014], Mansour et al. [2016]). Here, the incentive issues concern the learning algorithm itself, rather than the experts (or "arms") that it makes use of.

The question of how an expert should report beliefs has been studied before in the literature on strictly proper scoring rules [Brier, 1950, McCarthy, 1956, Savage, 1971, Gneiting and Raftery, 2007], but this literature typically considers the evaluation of a single prediction, rather than low-regret learning. Bayarri and DeGroot [1989] look at correlated settings where strictly proper scoring rules don't suffice, though they also do not consider how an aggregator can achieve low regret.

Finally, there are many works that fall under the broader umbrella of incentives in machine learning. Roughly, work in this area can be divided into two genres: incentives during the learning stage, e.g. [Cai et al., 2015, Shah and Zhou, 2015, Liu and Chen, 2016, Dekel et al., 2010], or incentives during the deployment stage, e.g. Brückner and Scheffer [2011], Hardt et al. [2016]. Finally, Babaioff et al. [2010] consider the problem of no-regret learning with selfish experts in an ad auction setting, where the incentives come from the allocations and payments of the auction, rather than from weights as in our case.

## 2 Preliminaries and Model

**Standard Model.** At each time step $t \in 1, ..., T$ we want to predict a binary realization $r^{(t)} \in \{0, 1\}$. To help in the prediction, we have access to $n$ experts that for each time step report a prediction $p_i^{(t)} \in [0, 1]$ about the realization. The realizations are determined by an oblivious adversary, and the predictions of the experts may or may not be accurate. The goal is to use the predictions of the experts in such a way that the algorithm performs nearly as well as the best expert in hindsight. Most of the algorithms proposed for this problem fall into the following framework.

**Definition 1** (Weight-update Online Prediction Algorithm)**.** A weight-update online prediction algorithm maintains a weight $w_i^{(t)}$ for each expert and makes its prediction $q^{(t)}$ based on $\sum_{i=1}^n w_i^{(t)} p_i^{(t)}$ and $\sum_i^n w_i^{(t)} (1 - p_i^{(t)})$. After the algorithm makes its prediction, the realization $r^{(t)}$ is revealed, and the algorithm updates the weights of experts using the rule

$$w_i^{(t+1)} = f\left(p_i^{(t)}, r^{(t)}\right) \cdot w_i^{(t)}, \tag{1}$$

where $f : [0, 1] \times \{0, 1\} \to \mathbb{R}^+$ is a positive function on its domain.

The standard WM algorithm has $f(p_i^{(t)}, r^{(t)}) = 1 - \eta \ell(p_i^{(t)}, r^{(t)})$ where $\eta \in (0, \frac{1}{2})$ is the learning rate, and predicts $q^{(t)} = 1$ if and only if $\sum_i^n w_i^{(t)} p_i^{(t)} \geq \sum_i^n w_i^{(t)} (1 - p_i^{(t)})$. Let the total loss of the algorithm be $M^{(T)} = \sum_{t=1}^T \ell(q^{(t)}, r^{(t)})$ and let the total loss of expert $i$ be $m_i^{(T)} = \sum_{t=1}^T \ell(p_i^{(t)}, r^{(t)})$. The MW algorithm has the property that $M^{(T)} \leq 2(1 + \eta) m_i^{(T)} + \frac{2 \ln n}{\eta}$ for each expert $i$, and RWM —where the algorithm picks 1 with probability proportional to $\sum_i^n w_i^{(t)} p_i^{(t)}$— satisfies $M^{(T)} \leq (1 + \eta) m_i^{(T)} + \frac{\ln n}{\eta}$ for each expert $i$ [Littlestone and Warmuth, 1994][Freund and Schapire, 1997]. The notion of *"no $\alpha$-regret"* [Kakade et al., 2009] captures the idea that the per time-step loss of an algorithm is $\alpha$ times that of the best expert in hindsight, plus a term that goes to 0 as $T$ grows:

**Definition 2** ($\alpha$-regret)**.** An algorithm is said to have *no $\alpha$-regret* if $M^{(T)} \leq \alpha \min_i m_i^{(T)} + o(T)$.

By taking $\eta = O(1/\sqrt{T})$, MW is a no 2-regret algorithm, and RWM is a no 1-regret algorithm.

**Selfish Model.** We consider a model in which experts have agency about the prediction they report, and care about the weight that they are assigned. In the selfish model, at time $t$ the expert formulates a private belief $b_i^{(t)}$ about the realization, but she is free to report any prediction $p_i^{(t)}$ to the algorithm. Let Bern$(p)$ be a Bernoulli random variable with parameter $p$. For any non-negative weight update function $f$,

$$\max_p \mathbb{E}_{b_i^{(t)}}[w_i^{(t+1)}] = \max_p \mathbb{E}_{r \sim \text{Bern}\left(b_i^{(t)}\right)}[f(p, r) w_i^{(t)}] = w_i^{(t)} \cdot \left(\max_p \mathbb{E}_{r \sim \text{Bern}\left(b_i^{(t)}\right)}[f(p, r)]\right).$$

So expert $i$ will report whichever $p_i^{(t)}$ will maximize the expectation of the weight update function.

Performance of an algorithm with respect to the reported loss of experts follows from the standard analysis [Littlestone and Warmuth, 1994]. However, the true loss may be worse (in Section 3 we

show this for the standard update rule, Section 4 shows it more generally). Unless explicitly stated otherwise, in the remainder of this paper $m_i^{(T)} = \sum_{t=1}^{T} \ell(b_i^{(t)}, r^{(t)})$ refers to the *true* loss of expert $i$. For now this motivates restricting the weight update rule $f$ to functions where reporting $p_i^{(t)} = b_i^{(t)}$ maximizes the expected weight of experts. We call these weight-update rules *Incentive Compatible (IC)*.

**Definition 3** (Incentive Compatibility). *A weight-update function $f$ is incentive compatible (IC) if reporting the true belief $b_i^{(t)}$ is always a best response for every expert at every time step. It is strictly IC when $p_i^{(t)} = b_i^{(t)}$ is the only best response.*

By a "best response," we mean an expected utility-maximizing report, where the expectation is with respect to the expert's beliefs.

*Collusion.* The definition of IC does not rule out the possibility that experts can collude to jointly misreport to improve their weights. We therefore also consider a stronger notion of incentive compatibility for groups with transferable utility.[8]

**Definition 4** (IC for Groups with Transferable Utility). *A weight-update function $f$ is IC for groups with transferable utility (TU-GIC) if for every subset $S$ of players, the total expected weight of the group $\sum_{i \in S} \mathbb{E}_{b_i^{(t)}}[w_i^{(t+1)}]$ is maximized by each reporting their private belief $b_i^{(t)}$.*

**Proper Scoring Rules.**   Incentivizing truthful reporting of beliefs has been studied extensively, and the set of functions that do this is called the set of proper scoring rules. Since we focus on predicting a binary event, we restrict our attention to this class of functions.

**Definition 5** (Binary Proper Scoring Rule, [Schervish, 1989]). *A function $f : [0, 1] \times \{0, 1\} \to \mathbb{R} \cup \{\pm\infty\}$ is a binary proper scoring rule if it is finite except possibly on its boundary and whenever for $p \in [0, 1]$ it holds that $p \in \max_{q \in [0,1]} p \cdot f(q, 1) + (1 - p) \cdot f(q, 0)$.*

A function $f$ is a *strictly* proper scoring rule if $p$ is the only value that maximizes the expectation. The first and perhaps most well-known proper scoring rule is the Brier scoring rule.

**Example 6** (Brier Scoring Rule, [Brier, 1950]). *The Brier score is $Br(p, r) = 2p_r - (p^2 + (1-p)^2)$ where $p_r = pr + (1-p)(1-r)$ is the report for the event that materialized.*

We will use the Brier scoring rule in Section 5 to construct an incentive-compatible randomized algorithm with good guarantees. The following proposition follows directly from Definitions 3 and 5.

**Proposition 7.** *Weight-update rule $f$ is (strictly) IC if and only if $f$ is a (strictly) proper scoring rule.*

Surprisingly, this result remains true even when experts can collude. While the realizations are obviously correlated, linearity of expectation causes the sum to be maximized exactly when each expert maximizes their own expected weight.

**Proposition 8.** *A weight-update rule $f$ is (strictly) incentive compatible for groups with transferable utility if and only if $f$ is a (strictly) proper scoring rule.*

Thus, for online prediction with selfish experts, we get TU-GIC "for free." It is quite uncommon for problems in non-cooperate game theory to admit good TU-GIC solutions. For example, results for auctions (either for revenue or welfare) break down once bidders collude, see e.g. [Goldberg and Hartline, 2005]. In the remainder of the paper we will simply use IC to refer to IC and TU-GIC, as strictly proper scoring rules yield algorithms that satisfy both definitions.

Thus, for IC algorithms we are restricted to considering (bounded) proper scoring rules as weight-update rules. Conversely, any bounded scoring rule can be used, possibly after an affine transformation (which preserve proper-ness). Are there any proper scoring rules that give an online prediction algorithm with a good performance guarantee? The standard algorithm for quadratic losses yields a weight-update function that is equivalent to the Brier strictly proper scoring rule, and thus is IC. The standard algorithm with absolute losses is not IC, so in the remainder of this paper we discuss this setting in more detail.

# 3 Deterministic Algorithms for Selfish Experts

This section studies the question if there are good online prediction algorithms with selfish experts for the absolute loss function. We restrict our attention here to deterministic algorithms; Section 5 gives a randomized algorithm with good guarantees.

Proposition 7 tells us that for selfish experts to have a strict incentive to report truthfully, the weight-update rule must be a strictly proper scoring rule. This section gives a deterministic algorithm based on the *spherical* strictly proper scoring rule that has no $(2 + \sqrt{2})$-regret (Theorem 10). Additionally, we consider the question if the non-truthful reports from experts in using the standard (non-IC) WM algorithm are harmful. We show that this is the case by proving it is not a no $(4 - O(1))$-regret algorithm for any constant smaller than $4$ (Proposition 11). This shows that, when experts are selfish, the IC online prediction algorithm with the spherical rule outperforms the standard WM algorithm (in the worst case).

**Online Prediction using a Spherical Rule.** We next give an algorithm that uses a strictly proper scoring rule that is based on the spherical rule scoring rule.[9] Consider the following weight-update rule:

$$f_{sp}\left(p_i^{(t)}, r^{(t)}\right) = 1 - \eta\left(1 - \left(1 - |p_i^{(t)} - r^{(t)}|\right)/\sqrt{p_i^{(t)} \cdot p_i^{(t)} + (1 - p_i^{(t)}) \cdot (1 - p_i^{(t)})}\right). \quad (2)$$

The following proposition establishes that this is in fact a strictly proper scoring rule. Due to space constraints, all proofs appear in Appendix A of the supplementary material.

**Proposition 9.** *The spherical weight-update rule in (2) is a strictly proper scoring rule.*

In addition to incentivizing truthful reporting, the WM algorithm with the update rule $f_{\text{sp}}$ does not do much worse than the best expert in hindsight.

**Theorem 10.** *WM with weight-update rule (2) for $\eta = O(1/\sqrt{T}) < \frac{1}{2}$ has no $(2 + \sqrt{2})$-regret.*

**True Loss of the Non-IC Standard Rule.** It is instructive to compare the guarantee in Theorem 10 with the performance of the standard (non-IC) WM algorithm. WM with the standard weight update function $f(p_i^{(t)}, r^{(t)}) = 1 - \eta|p_i^{(t)} - r^{(t)}|$ for $\eta \in (0, \frac{1}{2})$ has no 2-regret with respect to the *reported* loss of experts. However, this algorithm incentivizes extremal reports (for details see Appendix B in the supplementary material), and in the worst case, this algorithm's loss can be as bad as 4 times the *true* loss of the best expert in hindsight. Theorem 10 shows that a suitable IC algorithm obtains a superior worst-case guarantee.

**Proposition 11.** *The standard WM algorithm with weight-update rule $f\left(p_i^{(t)}, r^{(t)}\right) = 1 - \eta|p_i^{(t)} - r^{(t)}|$ results in a total worst-case loss no better than $M^{(T)} \geq 4 \cdot \min_i m_i^{(T)} - o(1)$.*

# 4 The Cost of Selfish Experts

We now address the third fundamental question: whether or not online prediction with selfish experts is strictly harder than with honest experts. As there exists a deterministic algorithm for honest experts with no 2-regret, showing a separation between honest and selfish experts boils down to proving that there exists a constant $\delta > 0$ such that best possible no $\alpha$-regret algorithm has $\alpha = 2 + \delta$. In this section we show that such a $\delta$ exists, and that it is independent of the learning rate. Hence the lower bound also holds for algorithms that, like the classical prediction algorithms, use a time-varying learning rate. Due to space considerations, this section only states the main results, for details and proofs refer to the supplementary materials where in Appendix D we give the results for IC algorithms, and in Appendix E we give the results for the non-IC algorithms. We extend these results to randomized algorithms in Section 5, where we rule out the existence of a (possibly randomized) no-regret algorithm for selfish experts.

**IC Algorithms.** To prove the lower bound, we have to be specific about which set of algorithms we consider. To cover algorithms that have a decreasing learning parameter, we first show that any positive proper scoring rule can be interpreted as having a learning parameter $\eta$.

**Proposition 12.** *Let $f$ be any strictly proper scoring rule. We can write $f$ as $f(p, r) = a + bf'(p, r)$ with $a \in \mathbb{R}$, $b \in \mathbb{R}^+$ and $f'$ a strictly proper scoring rule with $\min(f'(0, 1), f'(1, 0)) = 0$ and $\max(f'(0, 0), f'(1, 1)) = 1$.*

We call $f' : [0, 1] \times \{0, 1\} \to [0, 1]$ a *normalized* scoring rule. Using normalized scoring rules, we can define a family of scoring rules with different learning rates $\eta$. Define $\mathcal{F}$ as the following family of proper scoring rules generated by normalized strictly proper scoring rule $f$:

$$\mathcal{F} = \{f'(p, r) = a \left(1 + \eta(f(p, r) - 1)\right) : a > 0 \text{ and } \eta \in (0, 1)\}$$

By Proposition 12 the union of families generated by normalized strictly proper scoring rules cover all strictly proper scoring rules. Using this we can now formulate the class of deterministic algorithms that are incentive compatible.

**Definition 13** (Deterministic IC Algorithms). Let $\mathcal{A}_d$ be the set of deterministic algorithms that update weights by $w_i^{(t+1)} = a(1 + \eta(f(p_i^{(t)}, r^{(t)}) - 1))w_i^{(t)}$, for a normalized strictly proper scoring rule $f$ and $\eta \in (0, \frac{1}{2})$ with $\eta$ possibly decreasing over time. For $q = \sum_{i=1}^n w_i^{(t)} p_i^{(t)} / \sum_{i=1}^n w_i^{(t)}$, $A$ picks $q^{(t)} = 0$ if $q < \frac{1}{2}$, $q^{(t)} = 1$ if $q > \frac{1}{2}$ and uses any deterministic tie breaking rule for $q = \frac{1}{2}$.

Using this definition we can now state our main lower bound result for IC algorithms:

**Theorem 14.** *For the absolute loss function, there does not exists a deterministic and incentive-compatible algorithm $A \in \mathcal{A}_d$ with no 2-regret.*

Of particular interest are symmetric scoring rules, which occur often in practice, and which have a relevant parameter that drives the lower bound results:

**Definition 15** (Scoring Rule Gap). The *scoring rule gap* $\gamma$ of family $\mathcal{F}$ with generator $f$ is $\gamma = f(\frac{1}{2}) - \frac{1}{2}(f(0) + f(1)) = f(\frac{1}{2}) - \frac{1}{2}$.

By definition, the scoring rule gap for strictly proper scoring rules is strictly positive, and it drives the lower bound for symmetric functions:

**Lemma 16.** *Let $\mathcal{F}$ be a family of scoring rules generated by a symmetric strictly proper scoring rule $f$, and let $\gamma$ be the scoring rule gap of $\mathcal{F}$. In the worst case, MW with any scoring rule $f'$ from $\mathcal{F}$ with $\eta \in (0, \frac{1}{2})$ can do no better than $M^{(T)} \geq \left(2 + \frac{1}{\lceil \gamma^{-1} \rceil}\right) \cdot m_i^{(T)}$.*

As a consequence of Lemma 16, we can calculate lower bounds for specific strictly proper scoring rules. For example, the spherical rule used in Section 3 is a symmetric strictly proper scoring rule with a gap parameter $\gamma = \frac{\sqrt{2}}{2} - \frac{1}{2}$, and hence $1/\lceil \gamma^{-1} \rceil = \frac{1}{5}$.

**Non-IC Algorithms.** What about non-incentive-compatible algorithms? Could it be that, even with experts reporting strategically instead of honestly, there is a deterministic algorithm with loss at most twice that of the best expert in hindsight (or a randomized algorithm with vanishing regret), to match the classical results for honest experts? Under mild technical conditions, the answer is no. The following definition captures how players are incentivized to report differently from their beliefs.

**Definition 17** (Rationality Function). For a weight update function $f$, let $\rho_f : [0, 1] \to [0, 1]$ be the function from beliefs to predictions, such that reporting $\rho_f(b)$ is rational for an expert with belief $b$.

Under mild technical conditions on the rationality function, we show our main lower bound for (potentially non-IC) algorithms.

**Theorem 18.** *For a weight update function $f$ with continuous or non-strictly increasing rationality function $\rho_f$, there is no deterministic no 2-regret algorithm.*

Note that Theorem 18 covers the standard algorithm, as well as other common update rules such as the Hedge update rule $f_{\text{Hedge}}(p_i^{(t)}, r^{(t)}) = e^{-\eta|p_i^{(t)} - r^{(t)}|}$ [Freund and Schapire, 1997], and all IC methods, since they have the identity rationality function (though the bounds in Thm 14 are stronger).

# 5 Randomized Algorithms: Upper and Lower Bounds

**Impossibility of Vanishing Regret.** We now consider randomized online learning algorithms, which can typically achieve better worst-case guarantees than deterministic algoritms. For example, with honest experts, there are randomized algorithms no 1-regret. Unfortunately, the lower bounds in Section 4 imply that no such result is possible for randomized algorithms (more details in Appendix F).

**Corollary 19.** *Any incentive compatible randomized weight-update algorithm or non-IC randomized algorithm with continuous or non-strictly increasing rationality function cannot be no 1-regret.*

**An IC Randomized Algorithm.** While we cannot hope to achieve a no-regret algorithm for online prediction with selfish experts, we can do better than the deterministic algorithm from Section 3. Consider the following class of randomized algorithms:

**Definition 20** ($\theta$-randomized weighted majority). Let $\mathcal{A}_r$ be the class of algorithms that maintains expert weights as in Definition 1. Let $b^{(t)} = \sum_{i=1}^{n} \frac{w_i^{(t)}}{\sum_{j=1}^{n} w_j^{(t)}} \cdot p_i^{(t)}$ be the weighted predictions. For parameter $\theta \in [0, \frac{1}{2}]$ the algorithm chooses 1 with probability $p^{(t)} = \begin{cases} 0 & \text{if } b^{(t)} \leq \theta \\ b^{(t)} & \text{if } \theta < b^{(t)} \leq 1 - \theta \\ 1 & \text{otherwise} \end{cases}$.

We call algorithms in $\mathcal{A}_r$ $\theta$-RWM algorithms. We'll use the Brier rule $f_{\text{Br}}(p_i^{(t)}, r^{(t)}) = 1 - \eta((p_i^{(t)})^2 + (1 - p_i^{(t)})^2 + 1)/2 - (1 - s_i^{(t)}))$ with $s_i^{(t)} = |p_i^{(t)} - r^{(t)}|$.

**Theorem 21.** *Let $A \in \mathcal{A}_r$ be a $\theta$-RWM algorithm with the Brier weight update rule $f_{Br}$ and $\theta = 0.382$ and with $\eta = O(1/\sqrt{T}) \in (0, \frac{1}{2})$. A has no 2.62-regret.*

# 6 Simulations

The theoretical results presented so far indicate that when faced with selfish experts, one should use an IC weight update rule, and ones with smaller scoring rule gap are better. Two objections to these conclusions are: first, the presented results are *worst-case*, and may not represent behavior on a typical input. It is of particular interest to see if on non-worst-case inputs, the non-IC standard weight-update rule does better or worse than the IC methods proposed in this paper. Second, there is a gap between our upper and lower bounds for IC rules, so it's interesting to see what numerical regret is obtained.

**Results.** In our first simulation, experts are represented by a simple two-state hidden Markov model (HMM) with a "good" state and a "bad" state. Realization $r^{(t)}$ is given by a fair coin. For $r^{(t)} = 0$ (otherwise beliefs are reversed), in the good state expert $i$ believes $b_i^{(t)} \sim \min\{\text{Exp}(1)/5, 1\}$, in the bad state $b_i^{(t)} \sim \text{U}[\frac{1}{2}, 1]$. The probability to exit a state is $\frac{1}{10}$ for both states. This data generating process models that experts that have information about the event are more accurate than experts who lack the information. Figure 1a shows the regret as a function of time for the standard (non-IC) algorithm, and IC scoring rules including one from the Beta family [Buja et al., 2005] with $\alpha = \beta = \frac{1}{2}$. For the IC methods, experts report $p_i^{(t)} = b_i^{(t)}$, for the standard algorithm $p_i^{(t)} = 1$ if $b_i^{(t)} \geq \frac{1}{2}$ and $p_i^{(t)} = 0$ otherwise. The $y$ axis is the ratio of the total loss of each of the algorithms to the performance of the best expert at that time. The plot is for 10 experts, $T = 10,000$, $\eta = 10^{-2}$, and the randomized[10] versions of the algorithms, averaged over 30 runs. Varying model parameters and the deterministic version show similar results.

Each of the IC methods does significantly better than the standard weight-update algorithm, and even at $T = 200,000$ (not shown in the graph), the IC methods have a regret factor of about 1.003, whereas the standard algorithm still has 1.14. This gives credence to the notion that failing to account for incentive issues is problematic beyond the worst-case bounds presented earlier. Moreover, while there is a worst-case lower bound that rules out no-regret, for natural synthetic data, the loss of all the IC algorithms approaches that of the best expert in hindsight, while the standard algorithm fails to do

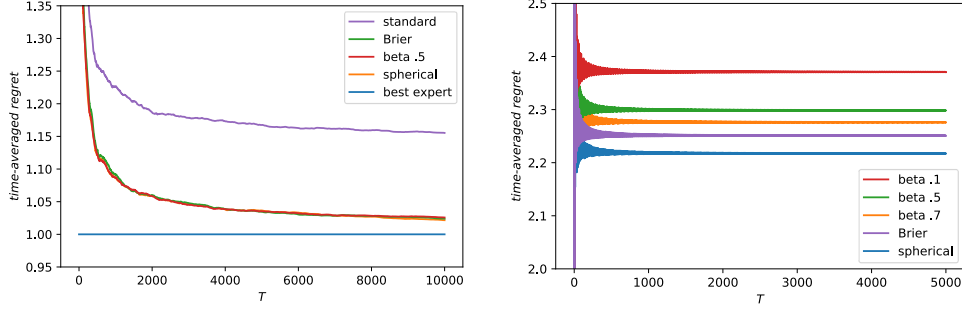

(a) The HMM data-generating process.  (b) The greedy lower bound instance.

Figure 1: Regret for different data-generating processes.

Table 1: Comparison of lower bound results with simulation. The simulation is run for $T = 10,000$, $\eta = 10^{-4}$ and we report the average of 30 runs. For the lower bounds, the first number is the lower bound from Lemma 16, i.e. $2 + \frac{1}{\lceil \gamma^{-1} \rceil}$, the second number (in parentheses) is $2 + \gamma$.

|  | Beta .1 | Beta .5 | Beta .7 | Beta .9 | Brier | Spherical |
|---|---|---|---|---|---|---|
| Greedy LB | 2.3708 | 2.2983 | 2.2758 | 2.2584 | 2.2507 | 2.2071 |
| LB Sim | 2.4414 | 2.3186 | 2.2847 | 2.2599 | 2.2502 | 2.2070 |
| Lem 16 LB | 2.33 (2.441) | 2.25 (2.318) | 2.25 (2.285) | 2.25 (2.260) | 2.25 | 2.2 (2.207) |

this. This seems to indicate that eliciting the truthful beliefs of the experts is more important than the exact weight-update rule.

*Comparison of LB Instances.* We consider both the lower bound instance described the proof of Lemma 16, and a greedy version that punishes the algorithm every time $w_0^{(t)}$ is "sufficiently" large.[11] Figure 1b shows the regret for different algorithms on the greedy lower bound instance. Table 1 shows that it very closely traces $2 + \gamma$, as do the numerical results for the lower bound from Lemma 16. In fact, for the analysis, we needed to use $\lceil \gamma^{-1} \rceil$ when determining the first phase of the instance. When we use $\gamma$ instead numerically, the regret seems to trace $2 + \gamma$ quite closely, rather than the weaker proven lower bound of $2 + \frac{1}{\lceil \gamma^{-1} \rceil}$. Table 1 shows that the analysis of Lemma 16 is essentially tight (up to the rounding of $\gamma$). Closing the gap between the lower and upper bound requires finding a different lower bound instance, or a better analysis for the upper bound.

## 7 Open Problems

There area number of interesting questions that this work raises. First of all, our utility model effectively causes experts to optimize their weight independently of other experts. Bayarri and DeGroot [1989] discuss different objective functions for experts, including optimizing relative weight among experts under different informational assumptions. These would impose different constraints as to which algorithms would lead to truthful reporting, and it would be interesting to see if no-regret learning is possible in this setting.

It also remains an open problem to close the gap between the best known upper and lower bounds that we presented in this paper. The simulations showed that the analysis for the lower bound instances is almost tight, so this requires a novel upper bound and/or a different lower bound instance.

Finally, strictly proper scoring rules are also well-defined beyond binary outcomes. It would be interesting to see what bounds can be proved for predictions over more than two outcomes.

## Footnotes

[1] https://fivethirtyeight.com/, https://www.nytimes.com/section/upshot.

[2] This is of course a simplification. FiveThirtyEight also uses features like the change in a poll over time, the state of the economy, and correlations between states. See https://fivethirtyeight.com/features/how-fivethirtyeight-calculates-pollster-ratings/ for details. Our goal in this paper is not to accurately model all of the fine details of FiveThirtyEight (which are anyways changing all the time). Rather, it is to formulate a general model of prediction with experts that clearly illustrates why incentives matter.

[3] https://projects.fivethirtyeight.com/pollster-ratings/

[4]More generally, one can investigate how the presence of machine learning algorithms affects data-generating processes, either during learning or deployment. We discuss some of this work in the related work section.

[5]When we speak of the best expert in hindsight, we are always referring to the true losses. Guarantees with respect to reported losses follow from standard results [Littlestone and Warmuth, 1994, Freund and Schapire, 1997, Cesa-Bianchi et al., 2007], but are not immediately meaningful.

[6]The loss function is often tied to the particular application. For example, in the current FiveThirtyEight pollster rankings, the performance of a pollster is primarily measured according to an absolute loss function and also whether the candidate with the highest polling numbers ended up winning (see https://github.com/fivethirtyeight/data/tree/master/pollster-ratings). However, in 2008 FiveThirtyEight used the notion of "pollster introduced error" or PIE, which is the square root of a difference of squares, as the most important feature in calculating the weights, see https://fivethirtyeight.com/features/pollster-ratings-v31/.

[7]The absolute loss function is known to elicit the median [Bonin, 1976] [Thomson, 1979], and since we have binary realizations, the median is either 0 or 1.

[8]Note that TU-GIC is a strictly stronger concept than IC and group IC with nontransferable utility (NTU-GIC) [Moulin, 1999][Jain and Mahdian, 2007].

[9]In Appendix G in the supplementary materials we give an intuition for why this rule yields better results than other natural candidates, such as the Brier scoring rule.

[10]Here we use the regular RWM algorithm, so in the notation of Section 5, we have $\theta = 0$.

[11]When $w_0^{(t)}$ is sufficiently large we make $e_0$ (and thus the algorithm) wrong twice: $b_0^{(t)} = 0$, $b_1^{(t)} = 1$, $b_2^{(t)} = \frac{1}{2}$, $r^{(t)} = 1$, and $b_0^{(t+1)} = 0$, $b_1^{(t)} = \frac{1}{2}$, $b_0^{(t)} = 1$, $r^{(t)} = 1$. "Sufficiently" here means that weight of $e_0$ is high enough for the algorithm to follow its advice during both steps.

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
