[Supplementary Material]

# A  Proofs

## A.1  Proof of Proposition 9

The spherical weight-update rule

$$f_{sp}\left(p_i^{(t)}, r^{(t)}\right) = 1 - \eta \left( 1 - \frac{1 - s_i^{(t)}}{\sqrt{\left(p_i^{(t)}\right)^2 + \left(1 - p_i^{(t)}\right)^2}} \right).$$

is a strictly proper scoring rule.

*Proof.* The standard spherical strictly proper scoring rule is $(1 - s_i^{(t)})/\sqrt{(p_i^{(t)})^2 + (1 - p_i^{(t)})^2}$. Any positive affine transformation of a strictly proper scoring rule yields another strictly proper scoring rule [Gneiting and Raftery, 2007], hence $(1 - s_i^{(t)})/\sqrt{(p_i^{(t)})^2 + (1 - p_i^{(t)})^2} - 1$ is also a strictly proper scoring rule. Now we multiply this by $\eta$ and add 1 to obtain

$$1 + \eta \left( \frac{1 - s_i^{(t)}}{\sqrt{\left(p_i^{(t)}\right)^2 + \left(1 - p_i^{(t)}\right)^2}} - 1 \right),$$

and rewriting proves the claim. □

## A.2  Proof of Theorem 10

WM with weight-update rule (2) for $\eta = O(1/\sqrt{T}) < \frac{1}{2}$ has no $(2 + \sqrt{2})$-regret.

*Proof.* We use an intermediate potential function $\Phi^{(t)} = \sum_i w_i^{(t)}$. Whenever the algorithm incurs a loss, the potential must decrease substantially. For the algorithm incur a loss, it must have picked the wrong outcome. Therefore it loss $|r^{(t)} - t^{(t)}| = 1$ and $\sum_i w_i^{(t)} s_i^{(t)} \geq \frac{1}{2} \cdot \Phi^{(t)}$. We use this to show that in those cases the potential drops significantly:

$$\Phi^{(t+1)} = \sum_i \left( 1 - \eta \left( 1 - \frac{1 - s_i^{(t)}}{\sqrt{\left(p_i^{(t)}\right)^2 + \left(1 - p_i^{(t)}\right)^2}} \right) \right) \cdot w_i^{(t)}$$

$$\leq \sum_i \left( 1 - \eta \left( 1 - \sqrt{2}\left(1 - s_i^{(t)}\right) \right) \right) \cdot w_i^{(t)} \qquad \left( \text{since } \min_x x^2 + (1-x)^2 = \frac{1}{2} \right)$$

$$= (1 - \eta)\,\Phi^{(t)} + \sqrt{2}\eta \sum_i \left( 1 - s_i^{(t)} \right) w_i^{(t)}$$

$$\leq (1 - \eta)\,\Phi^{(t)} + \frac{\sqrt{2}\eta}{2} \Phi^{(t)} \qquad \left( \text{since } \sum_i w_i^{(t)} \left( 1 - s_i^{(t)} \right) \leq \frac{1}{2}\Phi^{(t)} \right)$$

$$= \left( 1 - \frac{2 - \sqrt{2}}{2}\eta \right) \Phi^{(t)}$$

Since initially $\Phi^0 = n$, after $M^{(T)}$ mistakes, we have:

$$\Phi^T \leq n \left( 1 - \frac{2 - \sqrt{2}}{2}\eta \right)^{M^{(T)}}. \tag{3}$$

Now, let's bound the final weight of expert $i$ in terms of the number of mistakes she made:

$$w_i^{(T)} = \prod_t \left( 1 - \eta \left( 1 - \frac{1 - s_i^{(t)}}{\sqrt{\left(p_i^{(t)}\right)^2 + \left(1 - p_i^{(t)}\right)^2}} \right) \right)$$

$$\geq \prod_t \left( 1 - \eta s_i^{(t)} \right) \qquad \left( \text{since } \max_{x \in [0,1]} x^2 + (1-x)^2 = 1 \right)$$

$$\geq \prod_t (1 - \eta)^{s_i^{(t)}} \qquad (\text{since } 1 - \eta x \geq (1-\eta)^x \text{ for } x \in [0,1])$$

$$= (1 - \eta)^{\sum_t s_i^{(t)}}$$

$$= (1 - \eta)^{m_i^{(T)}}$$

Combining this with $w_i^{(t)} \leq \Phi^{(t)}$ and (3), and taking natural logarithms of both sides we get:

$$\ln\left( (1-\eta)^{m_i^{(T)}} \right) \leq \ln\left( n \left( 1 - \frac{2 - \sqrt{2}}{2}\eta \right)^{M^{(T)}} \right)$$

$$m_i^{(T)} \cdot \ln(1 - \eta) \leq M^{(T)} \cdot \ln\left( 1 - \frac{2 - \sqrt{2}}{2}\eta \right) + \ln n$$

$$m_i^{(T)} \cdot \left( -\eta - \eta^2 \right) \leq M^{(T)} \cdot \ln\left( \exp\left( -\frac{2 - \sqrt{2}}{2}\eta \right) \right) + \ln n$$

$$m_i^{(T)} \cdot \left( -\eta - \eta^2 \right) \leq M^{(T)} \cdot -\frac{2 - \sqrt{2}}{2}\eta + \ln n$$

$$M^{(T)} \leq \left( \frac{2}{2 - \sqrt{2}} \right) \cdot \left( (1 + \eta)m_i^{(T)} + \frac{\ln n}{\eta} \right)$$

where in the third inequality we used $-\eta - \eta^2 \leq \ln(1 - \eta)$ for $\eta \in (0, \frac{1}{2})$. Rewriting the last statement proves the claim. $\qquad\square$

### A.3  Proof of Proposition 11

The standard WM algorithm with weight-update rule $f\left( p_i^{(t)}, s_i^{(t)} \right) = 1 - \eta |p_i^{(t)} - r^{(t)}|$ results in a total worst-case loss no better than

$$M^{(T)} \geq 4 \cdot \min_i m_i^{(T)} - o(1).$$

*Proof.* Let $A$ be the standard weighted majority algorithm. We create an instance with 2 experts where $M^{(T)} \geq 4 \cdot \min_i m_i^{(T)} - o(1)$. Let the reports $p_1^{(t)} = 0$, and $p_2^{(t)} = 1$ for all $t \in 1, ..., T$; we will define $b_i^{(t)}$ shortly. Given the reports, $A$ will choose a sequence of predictions, let $r^{(t)}$ be 1 whenever the algorithm chooses 0 and vice versa, so that $M^{(T)} = T$.

Now for all $t$ such that $r^{(t)} = 1$, set $b_1^{(t)} = \frac{1}{2} - \epsilon$ and $b_2^{(t)} = 1$, and for all $t$ such that $r^{(t)} = 0$ set $b_1^{(t)} = 0$ and $b_2^{(t)} = \frac{1}{2} + \epsilon$, for small $\epsilon > 0$. Note that the beliefs $b_i^{(t)}$ indeed lead to the reports $p_i^{(t)}$ since $A$ incentivizes rounding the reports to the nearest integer.

Since the experts reported opposite outcomes, their combined total number of incorrect reports is $T$, hence the best expert had a reported loss of at most $T/2$. For each incorrect report $p_i^{(t)}$, the real loss of expert is $|r^{(t)} - b_i^{(t)}| = \frac{1}{2} + \epsilon$, hence $\min_i m_i^{(T)} \leq \left( \frac{1}{2} + \epsilon \right) T/2$, while $M^{(T)} = T$. Taking $\epsilon = o(T^{-1})$ yields the claim. $\qquad\square$

## A.4  Proofs of Proposition 12 and Theorem 14

These proofs appear in the text of Appendix D.

## A.5  Proof of Lemma 16

Let $\mathcal{F}$ be a family of scoring rules generated by a symmetric strictly proper scoring rule $f$, and let $\gamma$ be the scoring rule gap of $\mathcal{F}$. In the worst case, MW with any scoring rule from $\mathcal{F}$ with $\eta \in (0, \frac{1}{2})$ can do no better than

$$M^{(T)} \geq \left(2 + \frac{1}{\lceil \gamma^{-1} \rceil}\right) \cdot m_i^{(T)}.$$

*Proof.* Let $a, \eta$ be the parameters of $f'$ in the family $\mathcal{F}$, as in Definition 24. Fix $T$ sufficiently large and an integer multiple of $2\lceil \gamma^{-1} \rceil + 1$, and let $e_1$, $e_2$, and $e_3$ be three experts. For $t = 1, ..., \alpha \cdot T$ where $\alpha = \frac{2\lceil \gamma^{-1} \rceil}{2\lceil \gamma^{-1} \rceil + 1}$ such that $\alpha T$ is an even integer, let $p_1^{(t)} = \frac{1}{2}$, $p_2^{(t)} = 0$, and $p_3^{(t)} = 1$. Fix any tie-breaking rule for the algorithm. Realization $r^{(t)}$ is always the opposite of what the algorithm chooses.

Let $M^{(t)}$ be the loss of the algorithm up to time $t$, and let $m_i^{(t)}$ be the loss of expert $i$. We first show that at $t' = \alpha T$, $m_1^{(t')} = m_2^{(t')} = m_3^{(t')} = \frac{\alpha T}{2}$ and $M^{(t')} = \alpha T$. The latter part is obvious as $r^{(t)}$ is the opposite of what the algorithm chooses. That $m_1^{(t')} = \frac{\alpha T}{2}$ is also obvious as it adds a loss of $\frac{1}{2}$ at each time step. To show that $m_2^{(t')} = m_3^{(t')} = \frac{\alpha T}{2}$ we do induction on the number of time steps, in steps of two. The induction hypothesis is that after an even number of time steps, $m_2^{(t)} = m_3^{(t)}$ and that $w_2^{(t)} = w_3^{(t)}$. Initially, all weights are 1 and both experts have loss of 0, so the base case holds. Consider the algorithm after an even number $t$ time steps. Since $w_2^{(t)} = w_3^{(t)}$, $p_3^{(t)} = 1 - p_2^{(t)}$, and $p_1^{(t)} = \frac{1}{2}$ we have that $\sum_{i=1}^{3} w_i^{(t)} p_i^{(t)} = \sum_{i=1}^{3} w_i^{(t)}(1 - p_i^{(t)})$ and hence the algorithm will use its tie-breaking rule for its next decision. Thus, either $e_2$ or $e_3$ is wrong. Wlog let's say that $e_2$ was wrong (the other case is symmetric), so $m_2^{(t+1)} = 1 + m_3^{(t+1)}$. Now at time $t + 1$, $w_2^{(t+1)} = (1 - \eta)w_3^{(t+1)} < w_3^{(t+1)}$. Since $e_1$ does not express a preference, and $e_3$ has a higher weight than $e_2$, the algorithm will follow $e_3$'s advice. Since the realization $r^{(t+1)}$ is the opposite of the algorithms choice, this means that now $e_3$ incurs a loss of one. Thus $m_2^{(t+2)} = m_2^{(t+1)}$ and $w_2^{(t+2)} = w_2^{(t+1)}$ and $m_3^{(t+2)} = 1 + m_3^{(t+1)} = m_2^{(t+2)}$. The weight of expert $e_2$ is $w_2^{(t+2)} = aa(1 - \eta)w_2^{(t)}$ and the weight of expert $e_3$ is $w_3^{(t+2)} = a(1 - \eta)aw_3^{(t)}$. By the induction hypothesis $w_2^{(t)} = w_3^{(t)}$, hence $w_2^{(t+2)} = w_3^{(t+2)}$, and since we already showed that $m_2^{(t+2)} = m_3^{(t+2)}$, this completes the induction.

Now, for $t = \alpha T + 1, ..., T$, we let $p_1^{(t)} = 1$, $p_2^{(t)} = 0$, $p_3^{(t)} = \frac{1}{2}$ and $r^{(t)} = 0$. So henceforth $e_3$ does not provide information, $e_1$ is always wrong, and $e_2$ is always right. If we can show that the algorithm will always follow $e_1$, then the best expert is $e_2$ with a loss of $m_2^{(T)} = \frac{\alpha T}{2}$, while the algorithm has a loss of $M^{(T)} = T$. If this holds for $\alpha < 1$ this proves the claim. So what's left to prove is that the algorithm will always follow $e_1$. Note that since $p_3^{(t)} = \frac{1}{2}$ it contributes equal amounts to $\sum_{i=1}^{3} w_i^{(t)} p_i^{(t)}$ and $\sum_{i=1}^{3} w_i^{(t)}(1 - p_i^{(t)})$ and is therefore ignored by the algorithm in making its decision. So it suffices to look at $e_1$ and $e_2$. The algorithm will pick 1 iff $\sum_{i=1}^{3} w_i^{(t)}(1 - p_i^{(t)}) \leq \sum_{i=1}^{3} w_i^{(t)} p_i^{(t)}$, which after simplifying becomes $w_1^{(t)} > w_2^{(t)}$.

At time step $t$, $w_1^{(t)} = \left(a(1 + \eta(f(\frac{1}{2}) - 1))\right)^{\alpha T} \left(a \cdot (1 - \eta)\right)^{t - \alpha T}$ and $w_2^{(t)} = (a(1 - \eta))^{\frac{\alpha T}{2}} a^{\frac{\alpha T}{2} + t - \alpha T}$.

We have that $w_1^{(t)}$ is decreasing faster in $t$ than $w_2^{(t)}$. So if we can show that $w_1^{(T)} \geq w_2^{(T)}$ for some $\alpha < 1$, then $e_2$ will incur a total loss of $\alpha T/2$ while the algorithm incurs a loss of $T$ and the statement is proved.

We have that $w_1^{(t)}$ is decreasing faster in $t$ than $w_2^{(t)}$. So if we can show that at time $T$, $w_1^{(T)} \geq w_2^{(T)}$ for some $\alpha < 1$, then $e_2$ will incur a total loss of $\alpha T$ while the algorithm incurs a loss of $T$ and the statement is proved. First divide both weights by $a^T$ so that we have

$$a^{-T} w_1^{(T)} = \left(1 + \eta(f(\tfrac{1}{2}) - 1)\right)^{\alpha T} (1 - \eta)^{(1-\alpha)T}$$
$$a^{-T} w_2^{(T)} = (1 - \eta)^{\frac{\alpha T}{2}}.$$

Let $\alpha = \frac{2\lceil \gamma^{-1} \rceil}{2\lceil \gamma^{-1} \rceil + 1}$ and recall that $T = k \cdot \left(2\lceil \gamma^{-1} \rceil + 1\right)$ for positive integer $k$. Thus we can write

$$a^{-T} w_1^{(T)} = \left(1 + \eta(f(\tfrac{1}{2}) - 1)\right)^{k 2 \lceil \gamma^{-1} \rceil} (1 - \eta)^k$$
$$= \left(\left(1 + \eta(f(\tfrac{1}{2}) - 1)\right)^{2\lceil \gamma^{-1} \rceil} (1 - \eta)\right)^k$$
$$a^{-T} w_2^{(T)} = (1 - \eta)^{k \lceil \gamma^{-1} \rceil}$$
$$= \left((1 - \eta)^{\lceil \gamma^{-1} \rceil}\right)^k$$

So it holds that $w_1^{(T)} > w_2^{(T)}$ if we can show that $\left(1 + \eta(f(\tfrac{1}{2}) - 1)\right)^{2\lceil \gamma^{-1} \rceil} (1 - \eta) > (1 - \eta)^{\lceil \gamma^{-1} \rceil}$

$$
\begin{aligned}
\left(1 + \eta(f(\tfrac{1}{2}) - 1)\right)^{2\lceil \gamma^{-1} \rceil} (1 - \eta) &= \left(1 - (\tfrac{1}{2} - \gamma)\eta\right)^{2\lceil \gamma^{-1} \rceil} (1 - \eta) && \text{(def. of } \gamma \text{)} \\
&\geq (1 - \eta + 2\gamma\eta)^{\lceil \gamma^{-1} \rceil} (1 - \eta) && (4) \\
&= \left(\frac{1 - \eta + 2\gamma\eta}{1 - \eta}\right)^{\lceil \gamma^{-1} \rceil} (1 - \eta)^{\lceil \gamma^{-1} \rceil + 1} \\
&= (1 + 2\gamma\eta)^{\lceil \gamma^{-1} \rceil} (1 - \eta)^{\lceil \gamma^{-1} \rceil + 1} \\
&\geq \left(1 + \lceil \gamma^{-1} \rceil 2\gamma\eta\right) (1 - \eta)^{\lceil \gamma^{-1} \rceil + 1} \\
&\geq \left((1 + 2\eta)(1 - \eta)\right) (1 - \eta)^{\lceil \gamma^{-1} \rceil} \\
&> (1 - \eta)^{\lceil \gamma^{-1} \rceil} && \text{(for } \eta < \tfrac{1}{2} \text{)}
\end{aligned}
$$

Therefore expert $e_2$ will not incur any more loss during the last stage of the instance, so her total loss is $m_i^{(T)} = k\lceil \gamma^{-1} \rceil$ while the loss of the algorithm is $M^{(T)} = T = k \cdot \left(2\lceil \gamma^{-1} \rceil + 1\right)$. So

$$\frac{M^{(T)}}{m_i^{(t)}} \geq \frac{k \cdot \left(2\lceil \gamma^{-1} \rceil + 1\right)}{k\lceil \gamma^{-1} \rceil} = 2 + \frac{1}{\lceil \gamma^{-1} \rceil}$$

rearranging proves the claim. $\qquad \square$

### A.6 Proof of Theorem 18

The proof of this theorem appears in the text of Appendix E.

### A.7 Proof of Lemma 32

Let $\mathcal{F}$ be a family of scoring rules generated by a normalized strictly proper scoring rule $f$, with not both $f(0,0) = f(1,1)$ and $f(0,1) = f(1,0)$ and parameters $c$ and $d$ as in Definition 31. In the worst case, MW with any scoring rule $f'$ from $\mathcal{F}$ with $\eta \in (0, \tfrac{1}{2})$ can do no better than

$$M^{(T)} \geq \left(2 + \max\{\tfrac{1-c}{2c}, \tfrac{d}{4(1-d)}\}\right) \cdot m_i^{(T)}.$$

*Proof.* Fix $f$, and without loss of generality assume that $f(0,0) = 1$ (since $f$ is normalized, either $f(0,0)$ or $f(1,1)$ needs to be 1, rename if necessary). As $f$ is normalized, at least one of $f(0,1)$ and

$f(1,0)$ needs to be 0. For now, we consider the case where $f(0,1) = 0$, we treat the other case later. For now we have $f(0,0) = 1$, $f(0,1) = 0$, and by definition 31, $f(1,0) = 1-c$ and $f(1,1) = 1-d$, where $c > d$ (since correctly reporting 1 needs to score higher than reporting 0 when 1 materialized) and $\neg(c = 1 \wedge d = 0)$ (since that would put us in the semi-symmetric case).

We construct an instance as follows. We have two experts, $e_0$ reports 0 always, and $e_1$ reports 1 always, and as usual, the realizations are opposite of the algorithms decisions. Since the experts have completely opposite predictions, the algorithm will follow whichever expert has the highest weight. We will show that after a constant number of time steps $t$, the weight $w_0^{(t)}$ of $e_0$ will be larger than the weight $w_1^{(t)}$ of $e_1$ even though $e_0$ will have made one more mistake. Note that when this is true for some $t$ independent of $\eta$, this implies that the algorithm cannot do better than $2\frac{t}{t-1} > 2 + \frac{2}{t}$.

While it hasn't been the case that $w_0^{(t)} > w_1^{(t)}$ with $m_0^{(t)} = m_1^{(t)} + 1$, realizations alternate, and the weight of each expert is:

$$
\begin{aligned}
w_0^{(2t)} &= a^{2t}(1 + \eta(f(0,0) - 1))^t(1 + \eta(f(0,1) - 1))^t \\
&= a^{2t}(1 + \eta(1 - 1))^t(1 + \eta(1 - c - 1))^t \\
&= a^{2t}(1 - c\eta)^t \qquad\qquad\qquad\qquad\qquad\qquad (5) \\
w_1^{(2t)} &= a^{2t}(1 + \eta(f(1,1) - 1))^t(1 + \eta(f(1,0) - 1))^t \\
&= a^{2t}(1 + \eta((1 - d) - 1))^t(1 + \eta(0 - 1))^t \\
&= a^{2t}(1 - d\eta)^t(1 - \eta)^t \qquad\qquad\qquad\qquad\quad (6)
\end{aligned}
$$

What remains to be shown is that for some $t$ independent of $\eta$,
$$ a^{2t+1}(1 - c\eta)^{t+1} > a^{2t+1}(1 - d\eta)^{t+1}(1 - \eta)^t. $$

We know that it cannot be the case that simultaneously $c = 1$ and $d = 0$, so let's first consider the case where $c < 1$. In this case, it is sufficient to prove the above statement assuming $d = 0$, as this implies the inequality for all $d \in [0, c)$. The following derivation shows that $a^{2t+1}(1 - c\eta)^{t+1} > a^{2t+1}(1 - d\eta)^{t+1}(1 - \eta)^t$ whenever $\frac{c}{(1-c)} < t$.

$$
\begin{aligned}
a^{2t+1}(1 - c\eta)^{t+1} &> a^{2t+1}(1 - d\eta)^{t+1}(1 - \eta)^t \\
(1 - c\eta)^{t+1} &> (1 - \eta)^t \qquad\qquad\qquad\qquad\qquad (d = 0) \\
(1 - c\eta) &> \left(\frac{1 - \eta}{1 - c\eta}\right)^t \\
\ln(1 - c\eta) &> t \cdot \ln\left(\frac{1 - \eta}{1 - c\eta}\right) \\
1 - \frac{1}{1 - c\eta} &> t \cdot \left(\frac{1 - \eta}{1 - c\eta} - 1\right) \qquad\qquad (1 - \tfrac{1}{x} \leq \ln x \leq x - 1) \\
\frac{1 - c\eta - 1}{1 - c\eta} &> t \cdot \left(\frac{1 - \eta - 1 + c\eta}{1 - c\eta}\right) \\
\frac{c\eta}{1 - c\eta} &< t \cdot \frac{(1 - c)\eta}{1 - c\eta} \\
\frac{c\eta}{(1 - c)\eta} &< t \\
\frac{c}{(1 - c)} &< t
\end{aligned}
$$

So after $2t + 1$ time steps for some $t \leq \frac{c}{1-c} + 1$, expert $e_0$ will have one more mistake than expert $e_1$, but still have a higher weight. This means that after at most another $2t + 1$ time steps, she will

have two more mistakes, yet still a higher weight. In general, the total loss of the algorithm is at least $2 + \frac{1-c}{c}$ times that of the best expert. Now consider the case where $c = 1$ and therefore $d > 0$. We will show that after $2t + 1$ time steps for some $t \leq 2\frac{1-d}{d} + 1$ expert $e_0$ will have one more mistake than expert $e_1$.

$$
\begin{aligned}
a^{2t}(1 - c\eta)^{t+1} &> a^{2t}(1 - d\eta)^t(1 - \eta)^t(1 - d\eta) \\
(1 - \eta)^{t+1} &> (1 - d\eta)^{t+1}(1 - \eta)^t && (c = 1) \\
\frac{1 - \eta}{1 - d\eta} &> (1 - d\eta)^t \\
\ln\left(\frac{1 - \eta}{1 - d\eta}\right) &> t\ln(1 - d\eta) \\
1 - \frac{1 - d\eta}{1 - \eta} &> t(1 - d\eta - 1) && (1 - \tfrac{1}{x} \leq \ln x \leq x - 1) \\
\frac{1 - \eta - 1 + d\eta}{1 - \eta} &> -td\eta \\
\frac{(1 - d)\eta}{1 - \eta} &< td\eta \\
\frac{1 - d}{d}\frac{1}{1 - \eta} &< t \\
2\frac{1 - d}{d} &< t && (\text{by } \eta < \tfrac{1}{2})
\end{aligned}
$$

So in any case, after $t \leq 2\max\{\frac{c}{1-c}, \frac{1-d}{d}\} + 1$ time steps so the loss compared to the best expert is at least

$$
2 + \max\{\tfrac{1-c}{c}, \tfrac{d}{2(1-d)}\}.
$$

What remains to be proven is the case where $f(0,1) > 0$. In this case, it will have to be that $f(1,0) = 0$, as $f$ is normalized. And similarly to before, by Definition 31, we have $f(0,1) = 1 - c$ and $f(1,1) = 1 - d$ for $c > d$ and $\neg(c = 1 \wedge d = 0)$. Now, whenever $w_o^{(t)} > w_1(t)$, $e_0$ will predict 1 and $e_1$ predicts 0, and otherwise $e_0$ predicts 0 and $e_1$ predicts 1. As usual, the realizations are opposite of the algorithm's decisions. For now assume tie of the algorithm is broken in favor of $e_1$, then the weights will be identical to (5), (6). If the tie is broken in favor of $e_0$ initially, it takes at most twice as long before $e_0$ makes two mistakes in a row. Therefore, the loss with respect to the best expert in hindsight of an algorithm with any asymmetric strictly proper scoring rule is

$$
2 + \max\{\tfrac{1-c}{2c}, \tfrac{d}{4(1-d)}\}.
$$

$\square$

### A.8   Proof of Lemma 34

Let $f$ be a weight update function with a non-strictly increasing rationality function $\rho_f$, such that there exists $b_1 < b_2$ with $\rho_f(b_1) \geq \rho_f(b_2)$. For every deterministic algorithm, in the worst case

$$
M^{(T)} \geq (2 + |b_2 - b_1|)m_i^{(T)}.
$$

*Proof.* Fix $f$, $b_1$ and $b_2$ such that $\rho_f(b_1) \geq \rho_f(b_2)$ with $b_1 < b_2$. Let $\pi_1 = \rho_f(b_1)$, $\pi_2 = \rho_f(b_2)$, $b_0 = 1 - \frac{b_2 + b_1}{2}$, and $\pi_0 = \rho_f(b_0)$.

Let there be two experts $e_0$ and $e_1$. Expert $e_0$ always predicts $\pi_0$ with belief $b_0$. If $\pi_1 = \pi_2$, $e_1$ predicts $\pi_1$ (similar to Proposition 11, we first fix the predictions of $e_1$, and will give consistent beliefs later). Otherwise $\pi_1 > \pi_2$, and expert $e_1$ has the following beliefs (and corresponding predictions) at time $t$:

$$
b_1^{(t)} = \begin{cases} b_1 & \text{if } \frac{w_0^{(t)}\pi_0 + w_1^{(t)}\pi_2}{w_0^{(t)} + w_1^{(t)}} \geq \frac{1}{2} \\ b_2 & \text{otherwise} \end{cases}
$$

The realizations are opposite of the algorithm's decisions.

We now fix the beliefs of $e_1$ in the case that $\pi_1 = \pi_2$. Whenever $r^{(t)} = 1$, set expert $e_1$'s belief to $b_2$, and whenever $r^{(t)} = 0$, set her belief to $b_1$. Note that the beliefs indeed lead to the predictions she made, by the fact that $\pi_1 = \rho_f(b_1) = \rho_f(b_2)$.

For the case where $\pi_1 > \pi_2$, if $(w_0^{(t)}\pi_0 + w_1^{(t)}\pi_2)/(w_0^{(t)} + w_1^{(t)}) \geq \frac{1}{2}$ then $e_1$'s belief will be $b_1$ leading to a report of $\pi_1$ and as $\pi_1 > \pi_2$ it must hold that $(w_0^{(t)}\pi_0 + w_1^{(t)}\pi_1)(w_0^{(t)} + w_1^{(t)}) > \frac{1}{2}$, hence the algorithm will certainly choose 1, so the realization is 0. Conversely, if $(w_0^{(t)}\pi_0 + w_1^{(t)}\pi_2)(w_0^{(t)} + w_1^{(t)}) < \frac{1}{2}$, then the belief of $e_1$ will be $b_2$ and her report will lead the algorithm to certainly choose 0, so the realization is 1. So in all cases, if the realization is 1, then the belief of expert $e_1$ is $b_2$ and otherwise it is $b_1$.

The total number of mistakes $M^{(T)}$ for the algorithm after $T$ time steps is $T$ by definition. Every time the realization was 1, $e_0$ will incur loss of $\frac{b_1+b_2}{2}$ and $e_1$ incurs a loss of $1 - b_2$, for a total loss of $1 - b_2 + \frac{b_1+b_2}{2} = 1 - \frac{b_2-b_1}{2}$. Whenever the realization was 0, $e_0$ incurs a loss of $1 - \frac{b_1+b_2}{2}$ and $e_1$ incurs a loss of $b_1$ for a total loss of $1 - \frac{b_1+b_2}{2} + b_1 = 1 - \frac{b_2-b_1}{2}$.

So the total loss for *both* of the experts is $\left(1 - \frac{b_2-b_1}{2}\right) \cdot T$, so the best expert in hindsight has $m_i^{(T)} \leq \frac{1}{2}\left(1 - \frac{b_2-b_1}{2}\right) \cdot T$. Rewriting yields the claim. $\qquad \square$

## A.9  Proof of Theorem 21

Let $A \in \mathcal{A}$ be a $\theta$-RWM algorithm with the Brier weight update rule $f_{\mathrm{Br}}$ and $\theta = 0.382$ and with $\eta \in (0, \frac{1}{2})$. For any expert $i$ it holds that

$$M^{(T)} \leq 2.62 \left((1+\eta)m_i^{(T)} + \frac{\ln n}{\eta}\right).$$

*Proof.* The core difference between the proof of this statement, and the proof for Theorem 10 is in giving the upper bound of $\Phi^{(t+1)}$. Here we will give an upper bound of $\Phi^{(T)} \leq n \cdot \exp\left(-\frac{\eta}{2.62}M^{(T)}\right)$. Before giving this bound, observe that this would imply the theorem: since the weight updates are identical to the deterministic algorithm, we can use the same lower bound for $\Phi^{(T)}$, namely $\Phi^{(T)} \geq (1-\eta)^{m_i^{(T)}}$ for each expert. Then taking the log of both sides we get:

$$\ln n - \frac{\eta}{2.62}M^{(T)} \geq m_i^{(T)} \cdot \ln(1-\eta)$$
$$\ln n - \frac{\eta}{2.62}M^{(T)} \geq m_i^{(T)} \cdot (-\eta - \eta^2)$$
$$M^{(T)} \leq 2.62\left((1+\eta)m_i^{(T)} + \frac{\ln n}{\eta}\right)$$

So all that's left to prove is that whenever the algorithm incurs a loss $\ell$, $\Phi^{(t+1)} \leq \exp\left(-\frac{\eta}{2.62}\ell\right)$. At time $t$, the output $q^{(t)}$ of a $\theta$-RWM algorithm is one of three cases, depending on the weighted expert prediction. The first options is that the algorithm reported the realized event, in which case the $\ell^{(t)} = 0$ and the statement holds trivially. We treat the other two cases separately.

Let's first consider the case where the algorithm reported the incorrect event with certainty: $\ell^{(t)} = 1$. The means that $\sum_{i=1}^n w_i^{(t)}s_i^{(t)} \geq (1-\theta)\Phi^{(t)}$. Since the Brier rule is concave, $\Phi^{(t+1)}$ is maximized when $s_i^{(t)} = 1 - \theta$ for all experts $i$. In this case each we get

$$\Phi^{(t+1)} \leq \sum_i \left(1 - \eta \left(\frac{(p_i^{(t)})^2 + (1 - p_i^{(t)})^2 + 1}{2} - (1 - s_i^{(t)})\right)\right) w_i^{(t)}$$

$$\leq \sum_i \left(1 - \eta \left(\frac{(\theta)^2 + (1 - \theta)^2 + 1}{2} - \theta\right)\right) w_i^{(t)}$$

$$\leq \sum_i \left(1 - \frac{\eta}{2.62}\right) w_i^{(t)} \qquad \qquad \text{(since } \theta = .382)$$

$$= \left(1 - \frac{\eta}{2.62} \ell^{(t)}\right) \Phi^{(t)}.$$

Otherwise the algorithms report is between $\theta$ and $1 - \theta$. Let $\ell^{(t)} \in [\theta, 1 - \theta]$ be the loss of the algorithm. Again, since the Brier rule is concave, $\Phi^{(t+1)}$ is maximized when $s_i^{(t)} = \ell^{(t)}$ for all experts $i$. On $[\theta, 1 - \theta]$ the Brier proper scoring rule can be upper bounded by

$$1 - \frac{\eta}{f_{\mathrm{Br}}(1 - \theta, 1)/\theta} s_i^{(t)} = 1 - \frac{\eta}{2.62} s_i^{(t)}.$$

This yields

$$\Phi^{(t+1)} \leq \sum_i \left(1 - \eta \left(\frac{(p_i^{(t)})^2 + (1 - p_i^{(t)})^2 + 1}{2} - (1 - s_i^{(t)})\right)\right) w_i^{(t)}$$

$$\leq \sum_i \left(1 - \frac{\eta}{2.62} s_i^{(t)}\right) w_i^{(t)}$$

$$\leq \left(1 - \frac{\eta}{2.62} \ell^{(t)}\right) \Phi^{(t)}$$

So the potential at time $T$ can be bounded by $\Phi^{(T)} \leq n \cdot \prod_t \left(1 - \frac{\eta}{2.62} \ell^{(t)}\right) \leq n \cdot \exp\left(-\frac{\eta}{2.62} M^{(T)}\right)$, from which the claim follows. $\qquad \square$

## B  Incentive Compatibility of Standard Weighted Majority

When experts care about the weight that they are assigned, and with it their reputation and influence in the algorithm, different loss functions can lead to different expert behaviors. In the section B.2 we observe that for the quadratic loss function, in the standard WM and RWM algorithms, experts have no reason to misreport their beliefs. The next example shows that this is not the case for other loss functions, such as the absolute loss function.

### B.1  Absolute Losses

**Example 22.** Consider the standard WM algorithm, where each expert initially has unit weight, and an expert's weight is multiplied by $1 - \eta |p_i^{(t)} - r^{(t)}|$ at a time step $t$, where $\eta \in (0, \frac{1}{2})$ is the learning rate. Suppose there are two experts and $T = 1$, and that $b_1^{(1)} = .49$ while $b_2^{(1)} = 1$. Each expert reports to maximize her expected weight. Expanding, for each $i = 1, 2$ we have

$$\mathbb{E}[w_i^{(1)}] = \Pr(r^{(1)} = 1) \cdot (1 - \eta(1 - p_i^{(1)})) + \Pr(r^{(1)} = 0) \cdot (1 - \eta p_i^{(1)})$$

$$= b_i^{(1)} \cdot (1 - \eta(1 - p_i^{(1)})) + (1 - b_i^{(1)}) \cdot (1 - \eta p_i^{(1)})$$

$$= b_i^{(1)} - b_i^{(1)} \eta + b_i^{(1)} \eta p_i^{(1)} + 1 - \eta p_i^{(1)} - b_i^{(1)} + b_i^{(1)} \eta p_i^{(1)}$$

$$= 2 b_i^{(1)} \eta p_i^{(1)} - p_i^{(1)} \eta - b_i^{(1)} \eta + 1,$$

where all expectations and probabilities are with respect to the true beliefs of agent $i$. To maximize this expected weight over the possible reports $p_i^{(1)} \in [0, 1]$, we can omit the second two terms (which

are independent of $p_i^{(1)}$) and divide out by $\eta$ to obtain

$$\operatorname*{argmax}_{p_i^{(1)} \in [0,1]} 2b_i^{(1)}\eta p_i^{(1)} - p_i^{(1)}\eta - b_i^{(1)}\eta + 1 = \operatorname*{argmax}_{p_i^{(1)} \in [0,1]} p_i^{(1)}(2b_i^{(1)} - 1)$$

$$= \begin{cases} 1 & \text{if } b_i^{(1)} \geq \frac{1}{2} \\ 0 & \text{otherwise.} \end{cases}$$

Thus an expert always reports a point mass on whichever outcome she believes more likely. In our example, the second expert will report her true beliefs ($p_2^{(t)} = 1$) while the first will not ($p_1^{(t)} = 0$). While the combined true beliefs of the experts clearly favor outcome 1, the WM algorithm sees two opposing predictions and must break ties arbitrarily between them.

### B.2  Quadratic Losses

The first goal of this paper is to describe the class of algorithms that lead incentive compatible learning. Proposition 7 answers this question, so we can move over to our second goal, which is for different loss functions, do there exist incentive compatible algorithms with good performance guarantees? In this subsection we show that a corollary of Proposition 7 is that the standard MW algorithm with the quadratic loss function $\ell(p,r) = (p-r)^2$ is incentive compatible.

**Corollary 23.** *The standard WM algorithm with quadratic losses, i.e. $w_i^{(t+1)} = (1 - \eta(p_i^{(t)} - r^{(t)}))^2 \cdot w_i^{(t)}$ is incentive compatible.*

*Proof.* By Proposition 7 it is sufficient to show that $b_i^{(t)} = \max_p b_i^{(t)} \cdot (1 - \eta(p-1)^2) + (1 - b_i^{(t)}) \cdot (p-0)^2$.

$$\max_p b_i^{(t)} \cdot (1 - \eta(p-1)^2) + (1 - b_i^{(t)}) \cdot (1 - \eta(p-0)^2)$$
$$= \max_p b_i^{(t)} - b_i^{(t)}\eta p^2 + 2b_i^{(t)}\eta p - b_i^{(t)}\eta + 1 - b_i^{(t)} - \eta p^2 + b_i^{(t)}\eta p^2$$
$$= \max_p 1 - b_i^{(t)}\eta + 2b_i^{(t)}\eta p - \eta p^2$$
$$= \max_p 1 - b_i^{(t)}\eta + \eta p(2b_i^{(t)} - p)$$

To solve this for $p$, we take the derivative with respect to $p$: $\frac{d}{dp} 1 - b_i^{(t)}\eta + \eta p(2b_i^{(t)} - p) = \eta(2b_i^{(t)} - 2p)$. So the maximum expected value is uniquely $p = b_i^{(t)}$. $\qquad\square$

A different way of proving the Corollary is by showing that the standard update rule with quadratic losses can be translated into the Brier strictly proper scoring rule. Either way, for applications with quadratic losses, the standard algorithm already works out of the box. However, as we saw in Example 22, this is not the case with the absolute loss function. As the absolute loss function arises in practice—recall that FiveThirtyEight uses absolute loss to calculate their pollster ratings—in the remainder of this paper we focus on answering questions (2) and (3) from the introduction for the absolute loss function.

## C  Further Related Work

We survey in more detail the multiple other ways in which online learning and incentive issues have been blended, and the other efforts to model incentive issues in machine learning.

There is a large literature on prediction and decision markets (e.g. [Chen and Pennock, 2010, Horn et al., 2014]), which also aim to aggregate information over time from multiple parties and make use of proper scoring rules to do it. There are several major differences between our model and prediction markets. First, in our model, the goal is to predict a sequence of events, and there is feedback (i.e., the realization) after each one. In a prediction market, the goal is to aggregate information about a

single event, with feedback provided only at the end (subject to secondary objectives, like bounded loss).[12] Second, our goal is to make accurate predictions, while that of a prediction market is to aggregate information. For example, if one expert is consistently incorrect over time, we would like to ignore her reports rather than aggregate them with others' reports. Finally, while there are strong mathematical connections between cost function-based prediction markets and regularization-based online learning algorithms in the standard (non-IC) model [Abernethy et al., 2013], there does not appear to be analogous connections with online prediction with selfish experts.

There is also an emerging literature on "incentivizing exploration" (as opposed to exploitation) in partial feedback models such as the bandit model (e.g. [Frazier et al., 2014, Mansour et al., 2016]). Here, the incentive issues concern the learning algorithm itself, rather than the experts (or "arms") that it makes use of.

The question of how an expert should report beliefs has been studied before in the literature on strictly proper scoring rules [Brier, 1950, McCarthy, 1956, Savage, 1971, Gneiting and Raftery, 2007], but this literature typically considers the evaluation of a single prediction, rather than low-regret learning. The work by Bayarri and DeGroot [1989] specifically looks at the question of how an expert should respond to an aggregator who assigns and updates weights based on their predictions. Their work focuses on optimizing relative weight under different objectives and informational assumptions. However, it predates the work on low-regret learning, and it does not include performance guarantees for the aggregator over time. Boutilier [2012] discusses a model in which an aggregator wants to take a specific action based on predictions that she elicits from experts. He explores incentive issues where experts have a stake in the action that is taken by the decision maker.

Finally, there are many works that fall under the broader umbrella of incentives in machine learning. Roughly, work in this area can be divided into two genres: incentives during the learning stage, or incentives during the deployment stage. During the learning stage, one of the main considerations is incentivizing data providers to exert effort to generate high-quality data. There are several recent works that propose ways to elicit data in crowdsourcing applications in repeated settings through payments, e.g. [Cai et al., 2015, Shah and Zhou, 2015, Liu and Chen, 2016]. Outside of crowdsourcing, Dekel et al. [2010] consider a regression task where different experts have their own private data set, and they seek to influence the learner to learn a function such that the loss of their private data set with respect to the function is low.

During deployment, the concern is that the input is given by agents who have a stake in the result of the classification, e.g. an email spammer wishes to avoid its emails being classified as spam. Brückner and Scheffer [2011] model a learning task as a Stackelberg game. On the other hand Hardt et al. [2016] consider a cost to changing data, e.g. improving your credit score by opening more lines of credit, and give results with respect to different cost functions.

Online learning does not fall neatly into either learning or deployment, as the learning is happening while the system is deployed. Babaioff et al. [2010] consider the problem of no-regret learning with selfish experts in an ad auction setting, where the incentives come from the allocations and payments of the auction, rather than from weights as in our case.

# D  The Cost of Selfish Experts for IC Algorithms

We now address the third fundamental question: whether or not online prediction with selfish experts is strictly harder than with honest experts. In this section we restrict our attention to deterministic algorithms; we extend the results to randomized algorithms in Section F. As there exists a deterministic algorithm for honest experts where the loss is no more than twice that of the best expert in hindsight, showing a separation between honest and selfish experts boils down to proving that there exists a constant $\delta$ such that the worst-case loss is no better than a factor of $2 + \delta$ (with $\delta$ bounded away from 0 as $T \to \infty$). In this section we show that such a $\delta$ exists for all incentive compatible algorithms, and that $\delta$ depends on properties of a "normalized" version of the weight-update rule $f$, independent of the learning rate. This implies that the lower bound also holds for algorithms that, like the classical prediction algorithms, use a time-varying learning rate. In Section E we show that under mild technical conditions the true loss of non-IC algorithms is also bounded away from 2, and in Section F

the lower bounds for deterministic algorithms are used to show that there is no randomized algorithm that achieves vanishing regret.

To prove the lower bound, we have to be specific about which set of algorithms we consider. To cover algorithms that have a decreasing learning parameter, we first show that any positive proper scoring rule can be interpreted as having a learning parameter $\eta$.

**Proposition** (12). *Let $f$ be any strictly proper scoring rule. We can write $f$ as $f(p, r) = a + b f'(p, r)$ with $a \in \mathbb{R}$, $b \in \mathbb{R}^+$ and $f'$ a strictly proper scoring rule with $\min(f'(0, 1), f'(1, 0)) = 0$ and $\max(f'(0, 0), f'(1, 1)) = 1$.*

*Proof.* Let $f_{min} = \min(f(0, 1), f(1, 0))$ and $f_{max} = \max(f(0, 0), f(1, 1)) = 1$. Then define $f'(p, r) = \frac{f(p,r) - f_{min}}{f_{max} - f_{min}}$, $a = f_{min}$ and $b = f_{max} - f_{min}$. This is a positive affine translation, hence $f'$ is a strictly proper scoring rule. $\square$

We call $f' : [0, 1] \times \{0, 1\} \to [0, 1]$ a *normalized* scoring rule. Using normalized scoring rules, we can define a family of scoring rules with different learning rates $\eta$.

**Definition 24.** Let $f$ be any normalized strictly proper scoring rule. Define $\mathcal{F}$ as the following family of proper scoring rules generated by $f$:

$$\mathcal{F} = \{f'(p, r) = a\left(1 + \eta(f(p, r) - 1)\right) : a > 0 \text{ and } \eta \in (0, 1)\}$$

By Proposition 12 the union of families generated by normalized strictly proper scoring rules cover all strictly proper scoring rules. Using this we can now formulate the class of deterministic algorithms that are incentive compatible.

**Definition 25** (Deterministic Incentive-Compatible Algorithms). Let $\mathcal{A}_d$ be the set of deterministic algorithms that update weights by $w_i^{(t+1)} = a(1 + \eta(f(p_i^{(t)}, r^{(t)}) - 1))w_i^{(t)}$, for a normalized strictly proper scoring rule $f$ and $\eta \in (0, \frac{1}{2})$ with $\eta$ possibly decreasing over time. For $q = \sum_{i=1}^n w_i^{(t)} p_i^{(t)} / \sum_{i=1}^n w_i^{(t)}$, $A$ picks $q^{(t)} = 0$ if $q < \frac{1}{2}$, $q^{(t)} = 1$ if $q > \frac{1}{2}$ and uses any deterministic tie breaking rule for $q = \frac{1}{2}$.

Using this definition we can now state our main result:

**Theorem** (14). *For the absolute loss function, there does not exists a deterministic and incentive-compatible algorithm $A \in \mathcal{A}_d$ with no 2-regret.*

To prove Theorem 14 we proceed in two steps. First we consider strictly proper scoring rules that are symmetric with respect to the outcomes, because they lead to a lower bound that can be naturally interpreted by looking at the geometry of the scoring rule. We then extend these results to cover weight-update rules that use any (potentially asymmetric) strictly proper scoring rule.

### D.1 Symmetric Strictly Proper Scoring Rules

We first focus on symmetric scoring rules in the sense that $f(p, 0) = f(1 - p, 1)$ for all $p \in [0, 1]$. We can thus write these as $f(p) = f(p, 1) = f(1 - p, 0)$. Many common scoring rules are symmetric, including the Brier rule [Brier, 1950], the family of pseudo-spherical rules (e.g. [Gneiting and Raftery, 2007]), the power family (e.g. [Jose et al., 2008]), and the beta family [Buja et al., 2005] when $\alpha = \beta$. We start by defining the scoring rule gap for normalized scoring rules, which will determine the lower bound constant.

**Definition 26** (Scoring Rule Gap). The *scoring rule gap* $\gamma$ of family $\mathcal{F}$ with generator $f$ is $\gamma = f(\frac{1}{2}) - \frac{1}{2}(f(0) + f(1)) = f(\frac{1}{2}) - \frac{1}{2}$.

The following proposition shows that for all strictly proper scoring rules, the scoring rule gap must be strictly positive.

**Proposition 27.** *The scoring rule gap $\gamma$ of a family generated by a symmetric strictly proper scoring rule $f$ is strictly positive.*

*Proof.* Since $f$ is symmetric and a strictly proper scoring rule, we must have that $\frac{1}{2}f(\frac{1}{2}) + \frac{1}{2}f(\frac{1}{2}) > \frac{1}{2}f(0) + \frac{1}{2}f(1)$ (since an expert with belief $\frac{1}{2}$ must have a strict incentive to report $\frac{1}{2}$ instead of 1). The statement follows from rewriting. $\square$

We are now ready to prove our lower bound for all symmetric strictly proper scoring rules. The interesting case is where the learning rate $\eta \to 0$, as otherwise it is easy to prove a lower bound bounded away from 2.

The following lemma establishes that the gap parameter is important in proving lower bounds for IC online prediction algorithms. Intuitively, the lower bound instance exploits that experts who report $\frac{1}{2}$ will have a higher weight (due to the scoring rule gap) than an expert who is alternately right and wrong with extreme reports. This means that even though the indifferent expert has the same absolute loss, she will have a higher weight and this can lead the algorithm astray. The scoring rule gap is also relevant for the discussion in Appendix G. We give partial proof of the lemma below; the full proof appears in Appendix A.

**Lemma (16).** *Let $\mathcal{F}$ be a family of scoring rules generated by a symmetric strictly proper scoring rule $f$, and let $\gamma$ be the scoring rule gap of $\mathcal{F}$. In the worst case, MW with any scoring rule $f'$ from $\mathcal{F}$ with $\eta \in (0, \frac{1}{2})$ can do no better than*

$$M^{(T)} \geq \left(2 + \frac{1}{\lceil \gamma^{-1} \rceil}\right) \cdot m_i^{(T)}.$$

*Proof Sketch.* Let $a, \eta$ be the parameters of $f'$ in the family $\mathcal{F}$, as in Definition 24. Fix $T$ sufficiently large and an integer multiple of $2\lceil \gamma^{-1} \rceil + 1$, and let $e_1, e_2,$ and $e_3$ be three experts. For $t = 1, ..., \alpha \cdot T$ where $\alpha = \frac{2\lceil \gamma^{-1} \rceil}{2\lceil \gamma^{-1} \rceil + 1}$ such that $\alpha T$ is an even integer, let $p_1^{(t)} = \frac{1}{2}$, $p_2^{(t)} = 0$, and $p_3^{(t)} = 1$. Fix any tie-breaking rule for the algorithm. Realization $r^{(t)}$ is always the opposite of what the algorithm chooses.

Let $M^{(t)}$ be the loss of the algorithm up to time $t$, and let $m_i^{(t)}$ be the loss of expert $i$. We first show that at $t' = \alpha T$, $m_1^{(t')} = m_2^{(t')} = m_3^{(t')} = \frac{\alpha T}{2}$ and $M^{(t')} = \alpha T$. The latter part is obvious as $r^{(t)}$ is the opposite of what the algorithm chooses. That $m_1^{(t')} = \frac{\alpha T}{2}$ is also obvious as it adds a loss of $\frac{1}{2}$ at each time step. To show that $m_2^{(t')} = m_3^{(t')} = \frac{\alpha T}{2}$ we do induction on the number of time steps, in steps of two. The induction hypothesis is that after an even number of time steps, $m_2^{(t)} = m_3^{(t)}$ and that $w_2^{(t)} = w_3^{(t)}$. Initially, all weights are 1 and both experts have loss of 0, so the base case holds. Consider the algorithm after an even number $t$ time steps. Since $w_2^{(t)} = w_3^{(t)}$, $p_3^{(t)} = 1 - p_2^{(t)}$, and $p_1^{(t)} = \frac{1}{2}$ we have that $\sum_{i=1}^{3} w_i^{(t)} p_i^{(t)} = \sum_{i=1}^{3} w_i^{(t)} (1 - p_i^{(t)})$ and hence the algorithm will use its tie-breaking rule for its next decision. Thus, either $e_2$ or $e_3$ is wrong. Wlog lets say that $e_2$ was wrong (the other case is symmetric), so $m_2^{(t+1)} = 1 + m_3^{(t+1)}$. Now at time $t + 1$, $w_2^{(t+1)} = (1 - \eta)w_3^{(t+1)} < w_3^{(t+1)}$. Since $e_1$ does not express a preference, and $e_3$ has a higher weight than $e_2$, the algorithm will follow $e_3$'s advice. Since the realization $r^{(t+1)}$ is the opposite of the algorithms choice, this means that now $e_3$ incurs a loss of one. Thus $m_2^{(t+2)} = m_2^{(t+1)}$ and $w_2^{(t+2)} = w_2^{(t+1)}$ and $m_3^{(t+2)} = 1 + m_3^{(t+1)} = m_2^{(t+2)}$. The weight of expert $e_2$ is $w_2^{(t+2)} = aa(1 - \eta)w_2^{(t)}$ and the weight of expert $e_3$ is $w_3^{(t+2)} = a(1 - \eta)aw_3^{(t)}$. By the induction hypothesis $w_2^{(t)} = w_3^{(t)}$, hence $w_2^{(t+2)} = w_3^{(t+2)}$, and since we already showed that $m_2^{(t+2)} = m_3^{(t+2)}$, this completes the induction.

Now, for $t = \alpha T + 1, ..., T$, we let $p_1^{(t)} = 1$, $p_2^{(t)} = 0$, $p_3^{(t)} = \frac{1}{2}$ and $r^{(t)} = 0$. So henceforth $e_3$ does not provide information, $e_1$ is always wrong, and $e_2$ is always right. If we can show that the algorithm will always follow $e_1$, then the best expert is $e_2$ with a loss of $m_2^{(T)} = \frac{\alpha T}{2}$, while the algorithm has a loss of $M^{(T)} = T$. If this holds for $\alpha < 1$ this proves the claim. So what's left to prove is that the algorithm will always follow $e_1$. Note that since $p_3^{(t)} = \frac{1}{2}$ it contributes equal amounts to $\sum_{i=1}^{3} w_i^{(t)} p_i^{(t)}$ and $\sum_{i=1}^{3} w_i^{(t)} (1 - p_i^{(t)})$ and is therefore ignored by the algorithm in making its decision. So it suffices to look at $e_1$ and $e_2$. The algorithm will pick 1 iff $\sum_{i=1}^{3} w_i^{(t)} (1 - p_i^{(t)}) \leq \sum_{i=1}^{3} w_i^{(t)} p_i^{(t)}$, which after simplifying becomes $w_1^{(t)} > w_2^{(t)}$.

At time step $t$, $w_1^{(t)} = \left(a(1 + \eta(f(\frac{1}{2}) - 1))\right)^{\alpha T}(a \cdot (1 - \eta))^{t - \alpha T}$ and $w_2^{(t)} = (a(1 - \eta))^{\frac{\alpha T}{2}} a^{\frac{\alpha T}{2} + t - \alpha T}$.

We have that $w_1^{(t)}$ is decreasing faster in $t$ than $w_2^{(t)}$. So if we can show that $w_1^{(T)} \geq w_2^{(T)}$ for some $\alpha < 1$, then $e_2$ will incur a total loss of $\alpha T / 2$ while the algorithm incurs a loss of $T$ and the statement is proved. This is shown in the appendix. $\qquad\square$

As a consequence of Lemma 16, we can calculate lower bounds for specific strictly proper scoring rules. For example, the spherical rule used in Section 3 is a symmetric strictly proper scoring rule with a gap parameter $\gamma = \frac{\sqrt{2}}{2} - \frac{1}{2}$, and hence $1/\lceil \gamma^{-1} \rceil = \frac{1}{5}$.

**Corollary 28.** *In the worst case, the deterministic algorithm based on the spherical rule in Section 3 has*

$$M^{(T)} \geq \left(2 + \tfrac{1}{5}\right) m_i^{(T)}.$$

We revisit the scoring rule gap parameter again in Appendix G when we discuss considerations for selecting different scoring rules.

## D.2    Beyond Symmetric Strictly Proper Scoring Rules

We now extend the lower bound example to cover arbitrary strictly proper scoring rules. As in the previous subsection, we consider properties of normalized scoring rules to provide lower bounds that are independent of learning rate, but the properties in this subsection have a less natural interpretation.

For arbitrary strictly proper scoring rule $f'$, let $f$ be the corresponding normalized scoring rule, with parameters $a$ and $\eta$. Since $f$ is normalized, $\max\{f(0,0), f(1,1)\} = 1$ and $\min\{f(0,1), f(1,0)\} = 0$. We consider 2 cases, one in which $f(0,0) = f(1,1) = 1$ and $f(0,1) = f(1,0) = 0$ which is locally symmetric, and the case where at least one of those equalities does not hold.

**The semi-symmetric case.** If it is the case that $f$ has $f(0,0) = f(1,1) = 1$ and $f(0,1) = f(1,0) = 0$, then $f$ has enough symmetry to prove a variant of the lower bound instance discussed just before. Define the semi-symmetric scoring rule gap as follows.

**Definition 29** (Semi-symmetric Scoring Rule Gap). The *'semi-symmetric' scoring rule gap* $\mu$ of family $\mathcal{F}$ with normalized generator $f$ is $\mu = \frac{1}{2}\left(f(\frac{1}{2},0) + f(\frac{1}{2},1)\right) - \frac{1}{2}$.

Like the symmetric scoring rule gap, $\mu > 0$ by definition, as there needs to be a strict incentive to report $\frac{1}{2}$ for experts with $b_i^{(t)} = \frac{1}{2}$. Next, observe that since $f(\frac{1}{2},0), f(\frac{1}{2},1) \in [0,1]$ and $f(\frac{1}{2},0) + f(\frac{1}{2},1) = 1 + 2\mu$, it must be that $f(\frac{1}{2},0) \cdot f(\frac{1}{2},1) \geq 2\mu$. Using this it follows that:

$$
\begin{aligned}
&\left(1 + \eta(f(\tfrac{1}{2},0) - 1)\right)\left(1 + \eta(f(\tfrac{1}{2},1) - 1)\right) \\
&= 1 + \eta \cdot \left(f(\tfrac{1}{2},0) + f(\tfrac{1}{2},1) - 2\right) + \eta^2\left(f(\tfrac{1}{2},0) \cdot f(\tfrac{1}{2},1) - f(\tfrac{1}{2},0) - f(\tfrac{1}{2},1) + 1\right) \\
&= 1 + \eta \cdot (1 + 2\mu - 2) + \eta^2\left(f(\tfrac{1}{2},0) \cdot f(\tfrac{1}{2},1) - 2\mu\right) \\
&\geq 1 - \eta(1 - 2\mu) + \eta^2\left(2\mu - 2\mu\right) \\
&= 1 - \eta + 2\mu\eta
\end{aligned}
\tag{7}
$$

Now this can be used in the same way as we proved the setting before:

**Lemma 30.** *Let $\mathcal{F}$ be a family of scoring rules generated by a normalized strictly proper scoring rule $f$, with $f(0,0) = f(1,1)$ and $f(0,1) = f(1,0)$. In the worst case, MW with any scoring rule $f'$ from $\mathcal{F}$ with $\eta \in (0, \frac{1}{2})$ can do no better than*

$$M^{(T)} \geq \left(2 + \frac{1}{\lceil \mu^{-1} \rceil}\right) \cdot m_i^{(T)}.$$

*Proof Sketch.* Take the same instance as used in Lemma 16, with $\alpha = \frac{2\lceil \mu^{-1} \rceil}{2\lceil \mu^{-1} \rceil + 1}$. The progression of the algorithm up to $t = \alpha T$ is identical in this case, as expert $e_1$ is indifferent between outcomes, and $f(0,0) = f(1,1)$ and $f(0,1) = f(1,0)$ for experts $e_2$ and $e_3$. What remains to be shown is that the

weight of $e_1$ will be higher at time $T$. At time $T$ the weights of $e_1$ and $e_2$ are:

$$a^{-T} w_1^{(T)} = \left(1 + \eta(f(\tfrac{1}{2},0) - 1)\right)^{\frac{\alpha T}{2}} \left(1 + \eta(f(\tfrac{1}{2},1) - 1)\right)^{\frac{\alpha T}{2}} (1 - \eta)^{(1-\alpha)T}$$

$$a^{-T} w_2^{(T)} = (1 - \eta)^{\frac{\alpha T}{2}} .$$

Similarly to the symmetric case, wee know that $w_1^{(T)} > w_2^{(T)}$ if we can show that

$$\left(1 + \eta(f(\tfrac{1}{2},0) - 1)\right)^{\lceil \mu^{-1} \rceil} \left(1 + \eta(f(\tfrac{1}{2},1) - 1)\right)^{\lceil \mu^{-1} \rceil} (1 - \eta) > (1 - \eta)^{\lceil \mu^{-1} \rceil} .$$

By (7), it suffices to show that $(1 - \eta + 2\mu\eta)^{\lceil \mu^{-1} \rceil} (1-\eta) > (1 - \eta)^{\lceil \mu^{-1} \rceil}$, which holds by following the derivation in the proof of Lemma 16 given in the appendix, starting at (4). □

**The asymmetric case.** We finally consider the setting where the weight-update rule is not symmetric, nor is it symmetric evaluated only at the extreme reports. The lower bound that we show is based on the amount of asymmetry at these extreme points, and is parametrized as follows.

**Definition 31.** Let $c > d$ be parameters of a normalized strictly proper scoring rule $f$, such that $c = 1 - \max\{f(0,1), f(1,0)\}$ and $d = 1 - \min\{f(0,0), f(1,1)\}$.

Scoring rules that are not covered by Lemmas 16 or 30 must have either $c < 1$ or $d > 0$ or both. The intuition behind the lower bound instance is that two experts who have opposite predictions, and are alternatingly right and wrong, will end up with different weights, even though they have the same loss. We use this to show that eventually one expert will have a lower loss, but also a lower weight, so the algorithm will follow the other expert. This process can be repeated to get the bounds in the Lemma below. The proof of the lemma appears in the appendix.

**Lemma 32.** *Let $\mathcal{F}$ be a family of scoring rules generated by a normalized strictly proper scoring rule $f$, with not both $f(0,0) = f(1,1)$ and $f(0,1) = f(1,0)$ and parameters $c$ and $d$ as in Definition 31. In the worst case, MW with any scoring rule $f'$ from $\mathcal{F}$ with $\eta \in (0, \frac{1}{2})$ can do no better than*

$$M^{(T)} \geq \left(2 + \max\{\tfrac{1-c}{2c}, \tfrac{d}{4(1-d)}\}\right) \cdot m_i^{(T)}.$$

Theorem 14 now follows from combining the previous three lemmas.

*Proof of Theorem 14.* Follows from combining Lemmas 16, 30 and 32. □

# E The Cost of Selfish Experts for Non-IC Algorithms

What about non-incentive-compatible algorithms? Could it be that, even with experts reporting strategically instead of honestly, there is a deterministic no 2-regret algorithm (or a randomized algorithm with vanishing regret), to match the classical results for honest experts? Proposition 11 shows that the standard algorithm fails to achieve such a regret bound, but maybe some other non-IC algorithm does?

Typically, one would show that this is not the case by a "revelation principle" argument: if there exists some (non-IC) algorithm $A$ with good guarantees, then we can construct an algorithm $B$ which takes private values as input, and runs algorithm $A$ on whatever reports a self-interested agent would have provided to $A$. It does the strategic thinking for agents, and hence $B$ is an IC algorithm with the same performance as $A$. This means that generally, whatever performance is possible with non-IC algorithms can be achieved by IC algorithms as well, thus lower bounds for IC algorithms translate to lower bounds for non-IC algorithms. In our case however, the reports impact both the weights of experts as well as the decision of the algorithm simultaneously. Even if we insist on keeping the weights in $A$ and $B$ the same, the decisions of the algorithms may still be different. Therefore, rather than relying on a simulation argument, we give a direct proof that, under mild technical conditions, non-IC deterministic algorithms cannot be no 2-regret.[13] As in the previous section, we focus on

deterministic algorithms; Section F translates these lower bounds to randomized algorithms, where they imply that no vanishing-regret algorithms exist.

The following definition captures how players are incentivized to report differently from their beliefs.

**Definition 33** (Rationality Function). For a weight update function $f$, let $\rho_f : [0, 1] \to [0, 1]$ be the function from beliefs to predictions, such that reporting $\rho_f(b)$ is rational for an expert with belief $b$.

We restrict our attention here on rationality functions that are proper functions, meaning that each belief leads to a single prediction. Note that for incentive compatible weight update functions, the rationality function is simply the identity function.

In this section we show that for any deterministic prediction algorithm for which the rationality function is continuous or not strictly increasing, the worst-case loss of the algorithm is bounded away from twice the true loss of the best expert.[14] We start with a proof that any algorithm with non-strictly increasing rationality function must have worst-case loss strictly more than twice the best expert in hindsight. Conceptually, the proof is a generalization of the proof for Proposition 11.

**Lemma 34.** *Let $f$ be a weight update function with a non-strictly increasing rationality function $\rho_f$, such that there exists $b_1 < b_2$ with $\rho_f(b_1) \geq \rho_f(b_2)$. For every deterministic algorithm, in the worst case*

$$M^{(T)} \geq (2 + |b_2 - b_1|)m_i^{(T)}.$$

*Proof.* Fix, $f$, $b_1$ and $b_2$ such that $\rho_f(b_1) \geq \rho_f(b_2)$ with $b_1 < b_2$. Let $\pi_1 = \rho_f(b_1)$, $\pi_2 = \rho_f(b_2)$, $b_0 = 1 - \frac{b_2+b_1}{2}$, and $\pi_0 = \rho_f(b_0)$.

Let there be two experts $e_0$ and $e_1$. Expert $e_0$ always predicts $\pi_0$ with belief $b_0$. If $\pi_1 = \pi_2$, $e_1$ predicts $\pi_1$ (similar to Proposition 11, we first fix the predictions of $e_1$, and will give consistent beliefs later). Otherwise $\pi_1 > \pi_2$, and expert $e_1$ has the following beliefs (and corresponding predictions) at time $t$:

$$b_1^{(t)} = \begin{cases} b_1 & \text{if } \frac{w_0^{(t)}\pi_0 + w_1^{(t)}\pi_2}{w_0^{(t)} + w_1^{(t)}} \geq \frac{1}{2} \\ b_2 & \text{otherwise} \end{cases}$$

The realizations are opposite of the algorithm's decisions.

We now fix the beliefs of $e_1$ in the case that $\pi_1 = \pi_2$. Whenever $r^{(t)} = 1$, set expert $e_1$'s belief to $b_2$, and whenever $r^{(t)} = 0$, set her belief to $b_1$. Note that the beliefs indeed lead to the predictions she made, by the fact that $\pi_1 = \rho_f(b_1) = \rho_f(b_2)$.

For the case where $\pi_1 > \pi_2$, if $(w_0^{(t)}\pi_0 + w_1^{(t)}\pi_2)/(w_0^{(t)} + w_1^{(t)}) \geq \frac{1}{2}$ then $e_1$'s belief will be $b_1$ leading to a report of $\pi_1$ and as $\pi_1 > \pi_2$ it must hold that $(w_0^{(t)}\pi_0 + w_1^{(t)}\pi_1)(w_0^{(t)} + w_1^{(t)}) > \frac{1}{2}$, hence the algorithm will certainly choose 1, so the realization is 0. Conversely, if $(w_0^{(t)}\pi_0 + w_1^{(t)}\pi_2)(w_0^{(t)} + w_1^{(t)}) < \frac{1}{2}$, then the belief of $e_1$ will be $b_2$ and her report will lead the algorithm to certainly choose 0, so the realization is 1. So in all cases, if the realization is 1, then the belief of expert $e_1$ is $b_2$ and otherwise it is $b_1$.

The total number of mistakes $M^{(T)}$ for the algorithm after $T$ time steps is $T$ by definition. Every time the realization was 1, $e_0$ will incur loss of $\frac{b_1+b_2}{2}$ and $e_1$ incurs a loss of $1 - b_2$, for a total loss of $1 - b_2 + \frac{b_1+b_2}{2} = 1 - \frac{b_2-b_1}{2}$. Whenever the realization was 0, $e_0$ incurs a loss of $1 - \frac{b_1+b_2}{2}$ and $e_1$ incurs a loss of $b_1$ for a total loss of $1 - \frac{b_1+b_2}{2} + b_1 = 1 - \frac{b_2-b_1}{2}$.

So the total loss for *both* of the experts is $\left(1 - \frac{b_2-b_1}{2}\right) \cdot T$, so the best expert in hindsight has $m_i^{(T)} \leq \frac{1}{2}\left(1 - \frac{b_2-b_1}{2}\right) \cdot T$. Rewriting yields the claim. $\square$

For continuous rationality functions, we can generalize the results in Section D using a type of simulation argument. First, we address some edge cases.

**Proposition 35.** *For a weight update function $f$ with continuous strictly increasing rationality function $\rho_f$,*

1. *the regret is unbounded unless $\rho_f(0) < \frac{1}{2} < \rho(1)$; and*

2. *if $\rho_f(b) = \frac{1}{2}$ for $b \neq \frac{1}{2}$, the worst-case loss of the algorithm satisfies $M^{(T)} \geq (2 + |b - 1/2|) \, m_i^{(T)}$.*

*Proof.* First, assume that it does not hold that $\rho_f(0) < \frac{1}{2} < \rho_f(1)$. Since $\rho_f(0) < \rho_f(1)$ by virtue of $\rho_f$ being strictly increasing, it must be that either $\frac{1}{2} \leq \rho_f(0) < \rho_f(1)$ or $\rho_f(0) < \rho_f(1) \leq \frac{1}{2}$. Take two experts with $b_1^{(t)} = 0$ and $b_2^{(t)} = 1$. Realizations are opposite of the algorithm's predictions. Even though the experts have opposite beliefs, their predictions agree (potentially with one being indifferent), so the algorithm will consistently pick the same prediction, whereas one of the two experts will never make a mistake. Therefore the regret is unbounded.

As for the second statement. Since $\rho_f(0) < \frac{1}{2} < \rho_f(1)$, there is some $b$ such that $\rho_f(b) = \frac{1}{2}$. Wlog, assume $b < \frac{1}{2}$ (the other case is analogous). Since $\rho_f$ is continuous and strictly increasing, $\rho_f(\frac{b+1/2}{2}) > \frac{1}{2}$ while $\frac{b+1/2}{2} < \frac{1}{2}$. Take one expert $e_1$ with belief $b^{(t)} = \frac{b+1/2}{2} < \frac{1}{2}$, who will predict $p^{(t)} = \rho_f(\frac{b+1/2}{2}) > \frac{1}{2}$. Realizations are opposite of the algorithms decisions, and the algorithms decision is consistently 1, due to there only being one expert, and that expert putting more weight on 1. However, the absolute loss of the expert is only $\frac{1}{2} - \frac{|b-1/2|}{2}$ at each time step. Summing over the timesteps and rewriting yields the claim. $\qquad\square$

We are now ready to prove the main result in this section. The proof gives lower bound constants that are similar (though not identical) to the constants given in Lemmas 16, 30 and 32, though due to a reparameterization the factors are not immediately comparable.

**Theorem 36.** *For a weight update function $f$ with continuous strictly increasing rationality function $\rho_f$, with $\rho_f(0) < \frac{1}{2} < \rho_f(1)$ and $\rho_f(\frac{1}{2}) = \frac{1}{2}$, there is no deterministic no 2-regret algorithm.*

*Proof.* Fix $f$ with $\rho_f(0) < \frac{1}{2} < \rho_f(1)$ and $\rho_f(\frac{1}{2}) = \frac{1}{2}$. Define $p = \max\{\rho_f(0), 1 - \rho_f(1)\}$, so that $p$ and $1 - p$ are both in the image of $\rho_f$ and the difference between $p$ and $1 - p$ is as large as possible. Let $b_1 = \rho^{-1}(p)$ and $b_2 = \rho^{-1}(1 - p)$ and observe that $b_1 < \frac{1}{2} < b_2$.

Next, we rewrite the weight-update function $f$ in a similar way as the normalization procedure similar to Definition 24: $f(p,r) = a(1 + \eta(f'(p,r) - 1))$. where $\max\{f'(p,0), f'(1-p,1)\} = 1$ and $\min\{f'(p,1), f'(1-p,0)\} = 0$. Again we do this to prove bounds that are not dependent on any learning rate parameter.

Note that the composition of $\rho_f$ and $f$, namely $f(\rho_f(p), r)$ is a strictly proper scoring rule, since it changes the prediction in the same way as the selfish expert would do. Since $f(\rho_f(p), r)$, it must also be that $f'(\rho_f(p), r)$ is a strictly proper scoring rule, since it is a positive affine transformation of $f \circ \rho_f$.[15]

We now continue similarly to the lower bounds in Section D. We only treat the semi-symmetric and asymmetric cases as the former includes the special case of the symmetric weight-update function.

For the semi-symmetric case, by definition $f'(\rho_f(b_1), 0) = f'(\rho_f(b_2), 1) = 1$ and $f'(p,0), f'(1-p,1)\} = 1$ and $\min\{f'(p,1), f'(1-p,0) = 0$. Because $f' \circ \rho_f$ is a strictly proper scoring rule, the following inequality holds:

$$\tfrac{1}{2}f'(\rho_f(\tfrac{1}{2}), 0) + \tfrac{1}{2}f'(\rho_f(\tfrac{1}{2}), 1) + \mu = \tfrac{1}{2}f'(\rho_f(b_1), 0) + \tfrac{1}{2}f'(\rho_f(\tfrac{1}{2}), 1) = \tfrac{1}{2}$$

for some $\mu > 0$, since an expert with belief $\rho_f(\frac{1}{2})$ must have a strict incentive to report this. Here $\mu$ plays the same role as the semi-symmetric scoring rule gap.[16]

We now pitch three experts against each other in a similar lower bound instance as Lemma 30. For the first stage, they have beliefs $b_0^{(t)} = \frac{1}{2}$, $b_1^{(t)} = b_1$, $b_2^{(t)} = b_2$, so they have predictions $p_0^{(t)} = \frac{1}{2}$,

$p_1^{(t)} = \rho_f(b_1) = p$, $p_2^{(t)} = \rho_f(b_2) = 1 - p$. For the second stage, recall that either $b_1 = 0$ or $b_2 = 1$. In the former case, $b_0^{(t)} = 1$, $b_1^{(t)} = 0$, $b_2^{(t)} = \frac{1}{2}$ and $r^{(t)} = 0$ and in the latter case $b_0^{(t)} = 0$, $b_1^{(t)} = 1$, $b_2^{(t)} = \frac{1}{2}$ and $r^{(t)} = 1$. We now show a bijection between the instance in Lemma 30 and this instance, which establishes the lower bound for the semi-symmetric non-incentive compatible case. First of all, note that the weights of each of the experts in the first stage is the same (up to the choice of $a$ and $\eta$, and for now assuming that the algorithms choices and thus the realizations are the same):

$$w_0^{(2t)} = a^{2t}\left((1 + \eta(f'(\tfrac{1}{2}, 0) - 1))((1 + \eta(f'(\tfrac{1}{2}, 1) - 1))\right)^t$$
$$\geq a^{2t}(1 - \eta + 2\mu\eta)^t \qquad \text{(Follows from (7))}$$
$$w_1^{(2t)} = a^{2t}\left(1 - \eta)\right)^t$$
$$w_1^{(2t)} = a^{2t}\left(1 - \eta)\right)^t$$

In the second stage expert $e_0$ is always wrong and $e_1$ is always right, and hence at time $T$ the weights

Also note, that the predictions of $e_1$ and $e_2$ are opposite, i.e. $p$ and $1 - p$, so the algorithm will follow the expert which highest weight, meaning the algorithms decisions and the realizations are identical to the instance in Lemma 30.

To complete the proof of the lower bound instance, we need to show that the total loss of $e_1$ is the same. During the first stage, alternatingly the true absolute loss of $e_1$ is $b_1$ and $1 - b_1$, so after each 2 steps, her loss is 1. During the last stage, since her belief is certain (i.e. $b_0$ if $b_1 = 0$ or $b_2$ if $b_2 = 1$) ans she is correct, she incurs no additional loss. Therefore the loss of the algorithm and the true loss of $e_1$ are the same as in Lemma 30, hence the loss of the algorithm is at least $\frac{1}{\lceil\mu^{-1}\rceil}$ times that of the best expert in hindsight.

Finally, we consider the asymmetric case. We use a similar instance as Lemma 32 with two experts $e_0, e_1$. If $f'(1 - p, 0) = 0$ we have $b_0^{(t)} = b_1$ and $b_1^{(t)} = b_2$, so $p_0^{(t)} = p$ and $p_1^{(t)} = 1 - p$, otherwise the beliefs (and thus predictions) alternate. Again, the predictions are opposite of each other, and the weights evolve identically (up to the choice of $a$ and $\eta$) as before. Again the loss up until the moment that the same expert is chosen twice in a row is the same.

Once the same expert is chosen twice (after at most $2\max\{\frac{c}{1-c}, \frac{1-d}{d}\} + 1$) steps), it is not necessarily the case that the total loss of one expert exceeds the other by 2, as the true beliefs are $b_1$ and $b_2$, rather than 0 and 1. However, since at least either $b_1 = 0$ or $b_2 = 1$, and $b_1 < \frac{1}{2} < b_2$, the difference in total absolute loss in this non-IC instance is at least half of the IC instance, so we lose at most factor $\frac{1}{2}$ in the regret bound, hence for the asymmetric case $M^{(T)} \geq \left(2 + \max\{\frac{1-c}{4c}, \frac{d}{8(1-d)}\}\right)m_i^{(t)}$, completing the proof of the statement. $\qquad\square$

# F    Randomized Algorithms: Upper and Lower Bounds

## F.1    Impossibility of Vanishing Regret

We now consider randomized online learning algorithms, which can typically achieve better worst-case guarantees than deterministic algorithms. For example, with honest experts, there are randomized algorithms with worst-case loss $M^{(T)} \leq \left(1 + O\left(\frac{1}{\sqrt{T}}\right)\right)m_i^{(T)}$, which means that the regret with respect to the best expert in hindsight is vanishing as $T \to \infty$. Unfortunately, the lower bounds in Sections D and E below imply that no such result is possible for randomized algorithms.

**Corollary 37.** *Any incentive compatible randomized weight-update algorithm or non-IC randomized algorithm with continuous or non-strictly increasing rationality function cannot be no 1-regret.*

*Proof.* We can use the same instances as for Theorems 14 and 36 and Lemma 34 (whenever the algorithm was indifferent, the realizations were defined using the algorithm's tie-breaker rule; in the current setting simply pick any realization, say $r^t = 1$).

Whenever the algorithm made a mistake, it was because $\sum_i w_i^t s_i^t \geq \frac{1}{2}\sum_i w_i^t$. Therefore, in the randomized setting, it will still incur an expected loss of at least $\frac{1}{2}$. Therefore the total expected loss of the randomized algorithm is at least half that of the deterministic algorithm. Since the approximation

factor for the latter is bounded away from 2 in all cases in Theorems 14 and 36 and Lemma 34, in these cases the worst-case loss of a randomized algorithm satisfies $M^{(T)} \geq (1 + \Omega(1))m_i^{(T)}$. □

## F.2 An Incentive-Compatible Randomized Algorithm for Selfish Experts

While we cannot hope to achieve a no-regret algorithm for online prediction with selfish experts, we can do better than the deterministic algorithm from Section 3. We now focus on the more general class of algorithms where the algorithm can make any prediction $q^{(t)} \in [0, 1]$ and incurs a loss of $|q^{(t)} - r^{(t)}|$. We give a randomized algorithm based on the Brier strictly proper scoring rule with loss at most 2.62 times that of the best expert as $T \to \infty$.

Perhaps the most natural choice for a randomized algorithm is to simply report a prediction of $q^{(t)} = \sum_{i=1}^{n} w_i^{(t)} p_i^{(t)} / \sum_{j=1}^{n} w_j^{(t)}$. However, this is problematic when the experts are highly confident and correct in their predictions. By the definition of a (bounded) strictly proper scoring rule, $\frac{d}{dp_i^{(t)}} f(p_i^{(t)}, 1)$ is 0 at 1 (and similarly the derivative is 0 around 0 for a realization of 0). This means that experts that are almost certain and correct will not have their weight reduced meaningfully, and so the proof that uses the potential function does not go through.

This motivates looking for an algorithm where the sum of weights of experts is guaranteed to decrease significantly whenever the algorithm incurs a loss. Consider the following generalization of RWM that rounds predictions to the nearest integer if they are with $\theta$ of that integer.

**Definition 38** ($\theta$-randomized weighted majority). Let $\mathcal{A}_r$ be the class of algorithms that maintains expert weights as in Definition 1. Let $b^{(t)} = \sum_{i=1}^{n} \frac{w_i^{(t)}}{\sum_{j=1}^{n} w_j^{(t)}} \cdot p_i^{(t)}$ be the weighted predictions. For parameter $\theta \in [0, \frac{1}{2}]$ the algorithm chooses 1 with probability

$$
p^{(t)} = \begin{cases} 0 & \text{if } b^{(t)} \leq \theta \\ b^{(t)} & \text{if } \theta < b^{(t)} \leq 1 - \theta \\ 1 & \text{otherwise.} \end{cases}
$$

We call algorithms in $\mathcal{A}_r$ $\theta$-RWM algorithms. We'll use a $\theta$-RWM algorithm with the Brier rule. Recall that $s_i^{(t)} = |p_i^{(t)} - r^{(t)}|$; the Brier rule is defined as:

$$
f_{\text{Br}}(p_i^{(t)}, r^{(t)}) = 1 - \eta \left( \frac{(p_i^{(t)})^2 + (1 - p_i^{(t)})^2 + 1}{2} - (1 - s_i^{(t)}) \right). \tag{8}
$$

**Theorem 39.** *Let $A \in \mathcal{A}_r$ be a $\theta$-RWM algorithm with the Brier weight update rule $f_{Br}$ and $\theta = 0.382$ and with $\eta = O(1/\sqrt{T}) \in (0, \frac{1}{2})$. $A$ has no 2.62-regret.*

The proof appears in the appendix.

# G Selecting a Strictly Proper Scoring Rule

When selecting a strictly proper scoring rule for an IC online prediction algorithm, different choices may lead to very different guarantees. Many different scoring rules exist [McCarthy, 1956, Savage, 1971], and for discussion of selecting proper scoring rules in non-online settings, see also [Merkle and Steyvers, 2013]. Figure 2 shows two popular strictly proper scoring rules, the Brier rule and the spherical rule, along with the standard rule as comparison. Note that we have normalized all three rules for easy comparison.

Firstly, we know that for honest experts, the standard rule performs close to optimally. For every $\delta > 0$ we can pick a learning rate $\eta$ such that as $T \to \infty$ the loss of the algorithm $M^{(T)} \leq (2 + \delta)m_i^{(t)}$, while no algorithm could do better than $M^{(T)} < 2m_i^{(T)}$ [Littlestone and Warmuth, 1994, Freund and Schapire, 1997]. This motivates looking at strictly proper scoring rule that are "close" to the standard update rule in some sense. In Figure 2, if we compare the two strictly proper scoring rules, the spherical rule seems to follow the standard rule better than Brier does.

Figure 2: Three different normalized weight-update rules for $r^{(t)} = 1$. The line segment is the standard update rule, the concave curve the Brier rule and the other curve the spherical rule.

A more formal way of look at this is to look at the scoring rule gap. In Figure 2 we marked the $p = \frac{1}{2}$ location. Visually, the scoring rule gap $\gamma$ is the difference between a scoring rule and the standard rule at $p = \frac{1}{2}$. Since the Brier score has a large scoring rule gap, we're able to prove a strictly stronger lower bound for it: the scoring rule gap $\gamma = \frac{1}{4}$, hence MW with the Brier scoring rule cannot do better than $M^{(T)} \geq (2 + \frac{1}{4})m_i^{(T)}$ in the worst case, according to Lemma 16. Corollary 28 shows that for the Spherical rule, this factor is $2 + \frac{1}{5}$. The ability to prove stronger lower bounds for scoring rules with larger gap parameter $\gamma$ is an indication that it is probably harder to prove strong upper bounds for those scoring rules.

## Footnotes

[12]In the even more distantly related peer prediction scenario [Miller et al., 2005], there is never any realization at all.

[13]Similarly to Price of Anarchy (PoA) bounds, e.g. [Roughgarden and Tardos, 2007], the results here show the harm of selfish behavior. Unlike PoA bounds, our results are for dominant, though non-IC, strategies, rather than weaker equilibrium concepts such as the Nash equilibrium and our results are additive rather than multiplicative.

[14]This is true even when the learning rate is parameterized similarly to Definition 24, as the rationality function does not change for different learning rates due to the linearity of the expectation operator.

[15]And since $f'$ is a positive affine transformation of $f$, the rationality function is unchanged due to the linearity of the expectation operator.

[16]It is defined slightly differently though, as the image of $\rho_f$ may not be $[0, 1]$.