[Reviews · NeurIPS 2017]

Reviewer 1



This paper studies the problem of learning with expert advice when the experts are strategic. In particular, the authors focus on the study of (randomized) multiplicative weights algorithms and assume the experts aim to maximize their weights in the next round. The authors first show that the algorithm is incentive compatible if and only of the weight update function is a proper scoring rule. They then derive the regrets bounds with spherical scoring rules. The gap between the regret bounds between the setting with and without strategic experts are also derived. Overall, I like the paper. The technical results seem to be sound ( I didn’t carefully check all the proofs). While there are some assumptions I hope can be relaxed or better justified, I think the current results are interesting, and I would lean towards accepting the paper. Minor comments: - The connection between the proper scoring rule and weight update function is intuitive and interesting. The assumption that experts aim to maximize their weights is fine but seems a bit restrictive. I am wondering how easy it is to relax the assumption and extend it to, for example, the setting that experts aim to maximize some monotone function of the weights. - In Proposition 8, the authors show that they can get IC for groups with transferable utility for free. However, it might not be the case if the weights need to be normalized (e.g., the weights of all agents need to sum to 1). It seems to me this could be a more reasonable setting since experts might care about how well they are doing compared with other experts rather than just maximizing their own weights? - This is out of the scope of the paper: In the current paper, the experts already have the predictions, and they just need to decide what to report. I would be interested to see discussion on the setting when experts can exert different amounts of effort to obtain predictions with different qualities.

Reviewer 2



The paper studies the online prediction problem with "selfish" expert advice. In this setting, experts are allowed to make predictions other than their true beliefs in order to maximize their credibility. The contributions of the paper consist of three parts: 1) Definition of "incentive-compatible" (IC) algorithms which helps reasoning formally about the design and analysis of online prediciton algorithms. Establishing the connection between IC algorithms and Proper Scoring Rules. 2) Designing deterministic and random IC algorithms for absolute loss with non-trivial error bounds. 3) Establishing a hardness results for selfish setting comparing to honest setting via providing lower bounds for both IC and non-IC algorithms. The paper is written and organized well, and consequently, easy to read. The "selfish" setting of the paper is novel and well-motivated by real-world scenarios. The paper provides adequate and sound theoretical analysis in worst-case along with some experimental results illustrating the behavior of its introduced algorithms in typical data. The contributions of the paper seems theoretically significant.

Reviewer 3



In this paper, the standard (binary outcome) "prediction with expert advice" setting is altered by treating the experts themselves as players in the game. Specifically, restricting to the class of weight-update algorithms, the experts seek to maximize the weight assigned to them by the algorithm, as a notion of "rating" for their predictions. It is shown that learning algorithms are IC, meaning that experts reveal their true beliefs to the algorithm, if and only if the weight-update function is a strictly proper scoring rule. Because the weighted-majority algorithm (WM) has a weight update which is basically the negative of the loss function, WN is thus IC if and only if the (negative) loss is strictly proper. As the (negative) absolute loss function is not strictly proper, WM is not IC for absolute loss. The remainder of the paper investigates the absolute loss case in detail, showing that WM with the spherical scoring rule performs well, and moreover performs better than WM with absolute loss; no other deterministic IC (or even non-IC) algorithm for absolute loss can beat the performance in the non-strategic case; analogous results for the randomized case. I like the results, but find the motivation to be on loose footing. First, it is somewhat odd that experts would want to maximize the algorithm's weight assigned to them, as after all the algorithm's weights may not correspond to "performance" (a modified WM which publishes bogus weights intermittently can still achieve no regret), and even worse, the algorithm need not operate by calculating weights at all. I did find the 538 example compelling however. My biggest concern is the motivation behind scoring the algorithm or the experts with non-proper scoring rules. A major thrust of elicitation (see Making and Evaluating Point Forecasts, Gneiting 2011) is that one should always measure performance in an incentive-compatible way, which would rule out absolute loss. The authors do provide an example in 538's pollster ratings, but herein lies a subtlety: 538 is using absolute loss to score *poll* outcomes, based on the true election tallies. Poll results are not probabilistic forecasts, and in fact, that is a major criticism of polls from e.g. the prediction market community. Indeed, the "outcome" for 538's scoring is itself a real number between 0 and 1, and not just the binary 0,1 of e.g. whether the candidate won or lost. I would be much more positive about the paper if the authors could find a natural setting where non-proper losses would be used. Minor comment: Using "r" as the outcome may be confusing as it often denotes a report. It seems that "y" is more common, in both the scoring rules and prediction with expert advice literature.